# SCALABLE DO-SHAPLEY EXPLANATIONS WITH ESTIMAND-AGNOSTIC CAUSAL INFERENCE

## ABSTRACT

Among explainability techniques, SHAP stands out as one of the most popular, but often overlooks the causal structure of the problem. While do-SHAP uses interventional causal queries, its reliance on estimands hinders scalability. To address this problem, we propose employing estimand-agnostic Causal Inference, which allows for the estimation of any identifiable query with a single model, making do-SHAP feasible on arbitrarily complex graphs. We also develop a novel algorithm to significantly accelerate its computation at a negligible cost with a marked improvement in computational speed, as well as a method to explain inaccessible Data Generating Processes. We validate our approach on two real-world datasets, highlighting its potential in obtaining reliable explanations.

## 1 INTRODUCTION

The widespread adoption of Machine Learning (ML) systems has raised concerns about their limitations: models can replicate human biases (Angwin et al., 2016), base their outcomes on spurious correlations (Neuhaus et al., 2023), or be vulnerable to malicious adversarial attacks (Szegedy et al., 2014). Since most of these systems are black-boxes, there is an ever-increasing need for explainability techniques to make sense of the model. This is especially relevant *w.r.t.* fairness (i.e., protecting certain groups against discrimination), the right to explanation (European Commission, 2016) (e.g., "What would I need to do for my loan to be approved?"), debugging, auditing, and fostering user trust in the system.

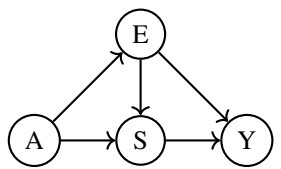

Figure 1: *Salary* causal graph.

As a response, the field of explainability has steadily gained traction, resulting in several approaches (Zhang et al., 2021) to explain model predictions. Among them, the Shapley value (SV, or SHAP) (Štrumbelj & Kononenko, 2014) is one of the most popular, since it is the only attribution strategy fulfilling a set of axioms that correspond to human intuition (see Appendix A). SVs are derived from a *value function* $\nu$, used to measure the effect of a subset (*coalition*) $\mathbf{S}$ of features $\mathbf{X}$ when applied on the model's prediction. Different definitions of $\nu$ result in different kinds of Shapley values, the most common being *marginal* and *conditional* SHAP (Chen et al., 2023).

However, both of these options ignore the *causal structure* underlying the data; for instance, Figure 1 represents the salary $Y$ of an employee of age $A$ with a certain education level $E$ and seniority level $S$. Let $f$ be a ML model $f(\mathbf{X}) \approx \mathbb{E}[Y \mid \mathbf{X}]$, learning $Y$ given inputs $\mathbf{X} = \{A, E, S\}$. Consider $\nu(\{E\})$. In marginal SHAP, $\nu$ assigns values $\{E = e\}$ and marginalizes the complementary set $(a, s) \sim P(A, S)$ regardless of how the coalition's values causally affect them $(E \to S)$. Conditional SHAP does consider these effects, but conditionally, $(a, s) \sim P(A, S \mid E = e)$, producing anti-causal effects to $A$ (i.e., we cannot change *age* by granting them a degree). Please refer to Appendix G for an extended discussion on this example.

Several works (Frye et al., 2020; Heskes et al., 2020; Lauritzen & Richardson, 2002; Janzing et al., 2020) discuss the limitations of non-causal SHAP and propose approaches with a causal interpretation under certain limitations, finally resulting in do-SHAP (Jung et al., 2022), whose value function is defined as $\nu(\mathbf{S}) = \mathbb{E}[f(\mathbf{X}) \mid do(\mathbf{S} = \mathbf{s})]$, where $do(\mathbf{S} = \mathbf{s})$ represents a causal intervention on $\mathbf{S}$ with values $\mathbf{s}$. Thanks to Causal Inference, we can transform this query into a probabilistic formula (the *estimand*) only containing terms from the observational distribution $P(\mathbf{X})$; hence, it is

possible to train ML models on each of these terms and apply them back into the formula to obtain an estimation for the query. The main drawback of *estimand-based* estimation is that do-SVs require computing $2^{|\mathbf{X}|}$ causal queries, one for each subset $\mathbf{S} \subseteq \mathbf{X}$, each with a different estimand formula and several ML models to estimate its terms, making it infeasible for complex graphs. In fact, do-SHAP's authors stated they "are not aware of any general causal effect estimators suitable for estimating the expression".

Here lies our first contribution: by employing the **estimand-agnostic** approach (Parafita & Vitrià, 2022) —based on Structural Causal Models (SCMs)—, we can compute any of the required causal effects with a single model. We do so by following a general procedure instead of query-specific estimands, thereby enabling the computation of do-SVs in a general, scalable way; see Sections 3.2 and 4.1. Secondly, we demonstrate new **do-SHAP properties** that result in the novel **Frontier-Reducibility Algorithm** (FRA), which optimizes do-SHAP significantly at virtually no cost; see Section 4.2. Next, we devise a explainability strategy with do-SVs, not only for accessible ML systems, but also for natural, *inaccessible* **Data Generating Processes**; see Section 4.3. Finally, we test the estimation capabilities of the estimand-agnostic approach on do-SVs, demonstrate the speedup of FRA, and showcase the application of these techniques on two real-world datasets to demonstrate the power of do-SHAP in providing reliable explanations; see Section 5 and Appendix F. We include an Impact Statement of our work in Appendix I.

## 2 RELATED WORK

Among many different kinds of explainability techniques (see (Zhang et al., 2021) for a recent survey), we are particularly interested in Shapley values (Lundberg & Lee, 2017), which provide an attribution for each input feature of the system to explain. There is a vast literature on SHAP estimation, discussing the choice of value function $\nu$ and tractable estimation strategies for the actual Shapley value (e.g., permutation sampling, adaptive sampling or model-specific strategies); refer to (Chen et al., 2023) for an extensive survey on the topic.

Our main interest is in SHAP approaches that leverage the underlying causal structure of the data. Asymmetric Shapley values (Frye et al., 2020) employ a topological order of the graph to restrict which permutations are considered in the computation of Conditional SHAP, thereby granting more attribution to ancestors of other features. Causal Shapley values (Heskes et al., 2020) properly considers the impact of causal interventions on Shapley attributions, but assumes a partial causal ordering of the graph to define causal chain graphs (Lauritzen & Richardson, 2002) in order to avoid the impact of causal confounders on identifiability. do-SHAP (Jung et al., 2022) does require a full causal graph, but provides a full method to compute attribution on all variables, as long as an estimand can be found for every causal query. Finally, in a different direction, Shapley flow (Wang et al., 2021) studies causal attributions on the causal graph's edges instead of its nodes/variables.

In order to avoid do-SHAP's scalability problems, we propose estimand-agnostic methods, which train SCMs modelling the data distribution and estimate causal queries. This approach is explored in the Neural Causal Models framework (Xia et al., 2021). In this line, recent contributions employ Deep Learning for SCM modeling: CausalGAN (Kocaoglu et al., 2018) uses Generative Adversarial Networks (Goodfellow et al., 2020) to model images in an SCM containing descriptive factors of the image; Parafita & Vitrià (Parafita & Vitrià, 2019) propose the Distributional Causal Node as a way to model mixed-type distributions (i.e., with discrete and continuous random variables) and expand their framework with Deep Causal Graphs (Parafita & Vitrià, 2022); Pawlowski *et al*. (Pawlowski et al., 2020) propose Normalizing Flows (Papamakarios et al., 2021) and Variational AutoEncoders (Kingma & Welling, 2014) for SCMs with medical image nodes; and Diffusion-based Causal Models (Chao et al., 2023) uses Diffusion Models (Ho et al., 2020) to train their SCMs on high-dimensional data. A promising alternative models SCMs not node by node as all previous works, but the graph as a whole with a single function of its noise signals, thereby avoiding error propagation when sampling. Two of these approaches are VACA (Sánchez-Martın et al., 2022), which uses Graph Neural Networks (Zhou et al., 2020), and Causal Normalizing Flows (Javaloy et al., 2024), with a single Normalizing Flow for the whole graph.

## 3 Preliminaries

This section establishes the concepts and notations needed in this work. We start with Structural Causal Models, an essential framework for causal queries, along with a discussion on the estimation of such queries. We then define the general Shapley value, from which we can derive do-SHAP.

**Notation** Sets, unless unambiguously differentiated, are represented by boldface letters ($\mathbf{X}$), while their elements are represented by simple letters ($X \in \mathbf{X}$). Let $\mathbb{P}(\mathbf{X}) := \{\varnothing \subseteq \mathbf{S} \subseteq \mathbf{X}\}$ denote the power set of $\mathbf{X}$, $[K] := \{1, \ldots, K\}$ an index set, $\Pi(\mathbf{S})$ the set of permutations of set $\mathbf{S}$ and $<_\pi$ the order entailed by $\pi$ (e.g., $3 <_\pi 2$ in $\pi = (3, 1, 2)$). Given a graph $\mathcal{G} = (\mathbf{V}, \mathbf{E})$ and a subset of vertices $\mathbf{X} \subseteq \mathbf{V}$, we denote the set of ancestors of $\mathbf{X}$ (including $\mathbf{X}$) as $An(\mathbf{X})$ and the set of descendants (including $\mathbf{X}$) as $De(\mathbf{X})$. For a certain node $X \in \mathbf{V}$, let $Pa_X$ denote the set of parents of $X$ (not including $X$). Random variables (r.v.s) are denoted in uppercase ($X$) with realizations in lowercase ($x$). Let $x \sim \mathcal{P}(X)$ denote the generation of a new sample $x$ from the distribution $P(X)$.

### 3.1 Structural Causal Models

Let $\mathcal{M} = (\mathcal{V}, \mathcal{W}, \mathcal{P}, \mathcal{F})$ be a Structural Causal Model (SCM), consisting of a set of *measured* r.v.s $\mathcal{V} = (V_1, \ldots, V_K)$, a set of *latent* r.v.s $\mathcal{W}$, their *priors* $\mathcal{P}(\mathcal{W}) = \prod_{W \in \mathcal{W}} \mathcal{P}(W)$ (all mutually independent), and a set of *functions* $\mathcal{F} := \{f_k\}_{k \in [K]}$ for each measured variable. The set of latent variables consists of $\mathcal{W} := \mathcal{E} \cup \mathcal{U}$, with $\mathcal{E} := (E_1, \ldots, E_K)$ the *exogenous noise signals*, $E_k$ corresponding to $V_k$, and a set of *confounders* $\mathcal{U} \subseteq \{U_{\{k,l\}} \mid 1 \leq k < l \leq K\}$[1], with each $U_{\{k,l\}} \in \mathcal{U}$ affecting both $V_k$ and $V_l$. Finally, each $f_k \in \mathcal{F}$ is a deterministic function $V_k = f_k(Pa_k, \mathcal{U}_{\{k,\cdot\}}, E_k)$ that returns $V_k$'s realizations given its measured parents $Pa_k \subseteq \mathcal{V} \setminus \{V_k\}$, confounders $\mathcal{U}_{\{k,\cdot\}} := \{U_{\{k,l\}}\}_{l \in [K]} \cap \mathcal{U}$ and the corresponding $E_k$. Let $Pa'_k := Pa_k \cup \mathcal{U}_{\{k,\cdot\}}$.

Let $\mathcal{G}_\mathcal{M} = (\mathbf{V}, \mathbf{E})$ be the *directed graph* associated to $\mathcal{M}$, with vertices, nodes or variables $\mathbf{V} := \mathcal{V} \cup \mathcal{W}$ and edges $\mathbf{E} := \{X \to V_k \mid \forall V_k \in \mathcal{V}, X \in Pa'_k \cup \{E_k\}\}$[2]. If $\mathcal{G}_\mathcal{M}$ is a *Directed Acyclic Graph* (DAG), there is a *topological order*[3] for $\mathcal{V}$. In that case, we can define $\mathcal{M}$'s probability distribution $\mathcal{P}_\mathcal{M}(\mathcal{V})$ from the $SCM$'s *sampling procedure*: it starts by sampling any latent variable $E_X \in \mathcal{E}, U \in \mathcal{U}$ from their priors $\varepsilon_X \sim \mathcal{P}(E_X), u \sim \mathcal{P}(U)$; then, following the topological order $k = 1..K$, it samples every $V_k \in \mathcal{V}$ by applying $v_k = f_k(pa_k, u_{\{k,\cdot\}}, \varepsilon_k)$.

We define an intervention $do(X = x)$, also denoted $\widehat{x}$, on a variable $X \in \mathcal{V}$ as the replacement of $f_x$ with the assignment $X \leftarrow x$. $X$ takes this value independently of its parents, and any descendants may be affected by this change. Note that this transforms the SCM $\mathcal{M}$ into an *intervened model* $\mathcal{M}_x$, graph $\mathcal{G}_x := \mathcal{G}_{\mathcal{M}_x}$ (without any edges pointing to $X$), and *distribution* $\mathcal{P}_x(\mathcal{V}) := \mathcal{P}_{\mathcal{M}_x}(\mathcal{V})$. We can also define interventions on sets of variables $do(\mathbf{X} = \mathbf{x})$ by the replacement of each of the corresponding functions $\{f_X \mid X \in \mathbf{X}\}$. The terms $P_\mathbf{x}(Y) = P(Y \mid do(\mathbf{X} = \mathbf{x})) = P(Y \mid \widehat{\mathbf{x}})$ are used interchangeably, for clarity or economy of notation depending on the case.

### 3.2 Identifiability and the estimand-agnostic approach

Let us assume a set of r.v.s $\mathcal{V}$ and an i. i. d. dataset $\mathcal{D} = (v^{(i)})_{i \in [N]} \sim \mathcal{P}(\mathcal{V})$ sampled from an unknown *Data Generating Process* (DGP) with a strictly positive probability measure $\mathcal{P}(\mathcal{V})$. Further assume that $\mathcal{P}(\mathcal{V})$ follows an unknown SCM $\mathcal{M}$, but whose graph $\mathcal{G}_\mathcal{M}$ is known. For instance, Figure 1 shows an SCM where $\mathcal{V} = (A, E, S, Y)$ and $\mathcal{U} = \varnothing$. Let us estimate the *causal query* $\mathcal{Q} := \mathbb{E}_Y[Y \mid \widehat{e}]$. Note that we can transform this interventional query into an observational formula —with no interventions— by leveraging do-calculus (Pearl, 2009) (see Appendix C), a set of operations to transform probabilistic expressions following their graph structure. At the end of this process, known as *identification*, we arrive at the final formula, the *estimand*: for this example, $\mathcal{Q} = \mathbb{E}_Y[Y \mid \widehat{e}] = \mathbb{E}_A[\mathbb{E}_Y[Y \mid e, A]]$. If such a formula exists, the query is said to be *identifiable* in that graph. Fortunately, there are algorithms to automatically determine identifiability and obtain the

---

[1] $\mathcal{U}$ can be empty, i.e., no latent confounders. This case is known as the *causal sufficiency assumption*.

[2] When representing $\mathcal{G}_\mathcal{M}$, we usually omit $\mathcal{W}$ as a notation shorthand; $\mathcal{E}$ is implicit, and confounders $U_{\{k,l\}}$ are denoted by $V_k \leftrightarrow V_l$.

[3] We say $\mathcal{V} = (V_1, \ldots, V_K)$ is in a *topological order* of the DAG $\mathcal{G}$ if $\forall k, l \in [K], V_k \in An(V_l) \Rightarrow k \leq l$. Let $<_\mathcal{G}$ represent the particular order defined by $\mathcal{G}$: $X <_\mathcal{G} Y$.

corresponding estimand (Shpitser & Pearl, 2006a;b), with implementations in R (Tikka & Karvanen, 2017) and Python (Pedemonte et al., 2021).

The estimand-based approach employs ML models to approximate each of the probabilistic terms in an estimand; in the example, we can train a classifier or regressor (depending on the kind of r.v. $Y$ is) to model $f(E, A) \approx \mathbb{E}_Y[Y \mid E, A]$ and then estimate the formula through Monte Carlo to arrive at an estimation for the query. However, this approach does not scale, since, for each and every query, we need to 1) derive the corresponding estimand for that query; 2) train ML models to estimate each term in the formula; and 3) put it all together to arrive at an answer for the query. Even with algorithms to automatically extract the estimand, it is not trivial to compute these formulas, especially if we need to answer exponentially many queries, as will be the case for do-SHAP.

However, if we had access to the original SCM $\mathcal{M}$, we could simply apply Monte Carlo by taking $N$ samples from the intervened distribution $P_e$, $(y^{(i)})_{i \in [N]} \sim \mathcal{P}_e(Y)$, using $\mathcal{M}_e$'s sampling procedure. Instead, let us consider a family of SCMs $\mathcal{M}_\Theta = (\mathcal{V}, \mathcal{W}, \mathcal{P}', \mathcal{F}_\Theta)$ with graph $\mathcal{G}_{\mathcal{M}_\Theta} = \mathcal{G}_\mathcal{M}$ and whose $\mathcal{F}_\Theta$ depends on a set of parameters $\Theta$ (models parameterized by $\Theta$). Irrespective of the choice of prior $\mathcal{P}'$ and functions $\mathcal{F}_\Theta$, if both are expressive enough, we can train $\mathcal{M}_\Theta$ to find a value $\theta$ so that the associated distribution $\mathcal{P}_{\mathcal{M}_\theta}(\mathcal{V}) = \mathcal{P}(\mathcal{V})$ (in an infinite data setting). If that is the case, by the application of procedures based on our *proxy SCM $\mathcal{M}_\theta$'s distribution*, *any* identifiable query can be estimated as if we were employing the underlying SCM $\mathcal{M}$, without ever using the estimand.

It is trivial to see why: since our identifiable query $Q$'s value is derived from the observational formula of the estimand, it depends exclusively on observational terms resulting from the joint distribution $\mathcal{P}_M(\mathcal{V})$, which we assume is correctly represented by our trained proxy $\mathcal{M}_\theta$. Therefore, as long as we derive its value from the distribution entailed by the proxy, we will necessarily arrive at the same result as with $\mathcal{M}$; otherwise, $\mathcal{P}_{\mathcal{M}_\theta}(\mathcal{V}) \neq \mathcal{P}_\mathcal{M}(\mathcal{V})$. In other words, even though its latent priors and functional assignments may be different, we can still compute the causal query through the proxy SCM *because* there is an estimand for $\mathcal{Q}$ in $\mathcal{G}_\mathcal{M}$. Hence, this results in an alternative approach for causal query estimation, the estimand-agnostic approach (Parafita & Vitrià, 2022): define a trainable SCM $\mathcal{M}'$ with the underlying SCM $\mathcal{G}$, train it to learn the observational distribution $\mathcal{P}_\mathcal{M}(\mathcal{V})$ and compute any identifiable queries from that single model using the SCM's procedures, not the specific estimand for each query. This will become essential for the computation of do-Shapley values.

### 3.3 THE SHAPLEY VALUE

Consider a set of $K$ players $\mathbf{X}$ and a *value function* $\nu : \mathbb{P}(\mathbf{X}) \to \mathbb{R}$. We can define the corresponding *coalitional* (cooperative) *game* $\Delta_\nu(\mathbf{S}) := \nu(\mathbf{S}) - \nu(\varnothing), \forall \mathbf{S} \in \mathbb{P}(\mathbf{X})$ such that $\Delta_\nu(\varnothing) = 0$. We define the Shapley value (Shapley, 1953) $\phi_{\Delta_\nu}(X)$ for a player $X \in \mathbf{X}$ (denoted by $\phi_\nu(X)$ or simply $\phi_X$ unless when leading to ambiguity) as:

$$\phi_{\Delta_\nu}(X) := \sum_{\mathbf{S} \subseteq \mathbf{X} \setminus \{X\}} \frac{1}{K} \binom{K-1}{|\mathbf{S}|}^{-1} (\nu(\mathbf{S} \cup \{X\}) - \nu(\mathbf{S})) \tag{1}$$

$$= \frac{1}{K!} \sum_{\pi \in \Pi(\mathbf{X})} (\nu(\mathbf{X}_{\leq_\pi X}) - \nu(\mathbf{X}_{<_\pi X})), \tag{2}$$

where $\mathbf{X}_{<_\pi X} := \{X' \in \mathbf{X} \mid X' <_\pi X\}$ and equivalently for $\mathbf{X}_{\leq_\pi X}$. Both equations are equivalent given that the sum over weighted subsets $\mathbf{S}$ results from the average over all permutations of the set of players $\mathbf{X}$. Note that SVs fulfill *efficiency*: $\sum_{X \in \mathbf{X}} \phi_X = \nu(\mathbf{X}) - \nu(\varnothing) = \Delta(\mathbf{X})$ (i.e., SHAP *attributions* add up to the contributions of the whole set $\mathbf{X}$).

### 3.4 TRACTABLE ESTIMATION OF THE SHAPLEY VALUE

Even though Equation (2) requires $2 \cdot K!$ computations of $\nu$, we can consider each permutation $\pi \in \Pi([K])$ as a sample from the uniform distribution over the set of permutations, $\pi \sim \mathcal{U}(\Pi([K]))$, resulting in $\phi_X = \mathbb{E}_{\pi \sim \mathcal{U}(\Pi([K]))}[\nu(\mathbf{X}_{\leq_\pi X}) - \nu(\mathbf{X}_{<_\pi X})]$, which can be approximated with Monte Carlo by sampling $N$ i.i.d. permutations and averaging their results (Mann & Shapley, 1960). This

estimator $\tilde{\phi}_X$ is unbiased *w.r.t.* $\phi_X$ with variance $\frac{\sigma_X^2}{N}$, where $\sigma_X^2$ is the variance of $\nu(\mathbf{X}_{\leq_\pi X}) - \nu(\mathbf{X}_{<_\pi X})$ over random $\pi$. Quasi-random and adaptive sampling strategies can also be employed for faster convergence of the Monte Carlo estimators; please refer to (Štrumbelj & Kononenko, 2014) for more details.

On the other hand, both methods result in a significant number of subset collisions, making it worthwhile to cache the $\nu(\mathbf{S})$ values to avoid unnecessary computations. We derive the expected number of coalitions sampled after $N$ permutations in Appendix B, which let us define a computational budget (i.e., how many permutations to sample) based on the desired coalition coverage.

## 4  METHOD

In the following, we present our contributions: we start defining the do-Shapley value (Jung et al., 2022) to propose estimand-agnostic techniques as a way to make do-SHAP feasible for arbitrary graphs; we derive several do-SHAP properties that motivate the definition of an efficient algorithm for a faster computation of do-SVs, the Frontier-Reducibility Algorithm; finally, we present a theorem allowing explanations on inaccessible DGPs. Please refer to Appendix H after reading this section for a detailed example illustrating the application of our approach.

### 4.1  THE DO-SHAPLEY VALUE

Consider an SCM $\mathcal{M} = (\mathcal{V}, \mathcal{W}, \mathcal{P}, \mathcal{F})$, a target r.v. $Y \in \mathcal{V}$, a subset of $K$ variables $\mathbf{X} \subseteq \mathcal{V} \setminus \{Y\}$ and a certain sample $\mathbf{x} \sim \mathcal{P}(\mathbf{X})$ we wish to explain. Given a coalition $\mathbf{S} \in \mathbb{P}(\mathbf{X})$ with realizations $\mathbf{s}$ (a subset of $\mathbf{x}$), let us define the *value function* $\nu_\mathbf{x}(\mathbf{S})$:

$$\nu_\mathbf{x}(\mathbf{S}) := \mathbb{E}\left[Y \mid do(\mathbf{S} = \mathbf{s})\right] \tag{3}$$

Then, the do-Shapley value (do-SV) (Jung et al., 2022) over variables $\mathbf{X}$ with realizations $\mathbf{x} \sim \mathcal{P}(\mathbf{X})$ on a variable $X \in \mathbf{X}$ is $\phi_X := \phi_{\nu_\mathbf{x}}(X)$. For economy of notation, we will simply write $\nu := \nu_\mathbf{x}$.

**Assumption 4.1.** We assume $\mathcal{P}(\mathcal{V})$ to be strictly positive, resulting from an unknown SCM $\mathcal{M}$, but whose graph $\mathcal{G}_\mathcal{M}$ is known[4], a DAG, and s.t. its do-SVs are *identifiable* in $\mathcal{G}_\mathcal{M}$ (i.e., all $\nu(\mathbf{S})$ terms, with $\mathbf{S} \subseteq \mathbf{X}$, are identifiable[5]). Note that $\mathcal{G}_\mathcal{M}$ may include latent confounders as long as its do-SVs are identifiable.

Jung et al. (2022) employed the estimand-based approach, which requires an estimand for each of the $2^{|\mathbf{X}|}$ terms $\nu(\mathbf{S})$. This makes do-SHAP impractical, since each estimand requires different ML models for its probability terms and an ad-hoc computation following the estimand formula. In response, we propose the estimand-agnostic approach: 1) train a single SCM to learn $\mathcal{P}(\mathcal{V})$; 2) for each query $\nu(\mathbf{S})$, determine if it is identifiable (as we do in the estimand-based approach); and 3) use general SCM-based procedures, not the estimand, to estimate the query. We do not include further details about SCM modeling and query estimation in this work because it is already covered in the respective papers (e.g., (Kocaoglu et al., 2018; Pawlowski et al., 2020; Javaloy et al., 2024)), which would unduly expand our already lengthy appendix. For a detailed explanation on a general framework for SCM training and estimation procedures, please refer to (Parafita & Vitrià, 2022).

### 4.2  EFFICIENT ESTIMATION OF THE DO-SHAPLEY VALUE

In this section, we derive several do-SHAP properties that will motivate a novel algorithm to accelerate its computation. We leave the proofs and derivation of the algorithm to Appendix D.

---

[4]This is a standard assumption in the Structural Causal Models community. If the graph is not available, Causal Discovery algorithms (Spirtes & Zhang, 2016) can be employed.

[5]Running the identifiability algorithms (Section 3.2) on all $2^{|\mathbf{X}|}$ terms a priori is unnecessary. Instead, when using the approximation method discussed in Section 3.4, we can test identifiability for each new sampled query, caching results for repeated coalitions. If any coalition is found to be non-identifiable during this process, an error state should halt it immediately; otherwise, if no non-identifiable coalition is found, our do-SV estimation will be valid. Moreover, certain graph structures (e.g., no latent confounders) make do-SVs trivially identifiable; a general graphical criterion for do-SV identifiability is left for future work.

**Proposition 4.2.** *For any non-ancestor $X$ of $Y$, $\phi_X = 0$.*

Consequently, we can restrict $\mathcal{G}$ to $Y$'s ancestors, since every other do-SV will necessarily be 0.

**Assumption 4.3.** Given an SCM $\mathcal{M} = (\mathcal{V}, \mathcal{W}, \mathcal{P}, \mathcal{F})$ and a target r.v. $Y \in \mathcal{V}$, we assume $\mathcal{M}$ to be the projected SCM $\mathcal{M}[An(Y)]$ (see Definition C.8) and simply denote it $\mathcal{M}$. From now on, $\mathcal{V} = \mathbf{X} \cup \{Y\}$ with $\mathbf{X} := An(Y) \setminus \{Y\} = (V_0, \ldots, V_{K-1})$ in a topological order. Let $Y := V_K$.

Next, we introduce the concept of *frontiers*, with which we derive essential properties necessary to define an algorithm to speed up do-SHAP. Please refer to Appendix D.2 for a complete demonstration of these properties and the derivation of Algorithm 1.

**Definition 4.4.** Given any node $X \in \mathbf{X}$, a subset $\mathbf{S} \subseteq \mathbf{X}$ is a frontier between $X$ and $Y$ if $X \notin \mathbf{S}$ and all directed paths $p = (X, \ldots, Y)$ from $X$ to $Y$ are blocked by $\mathbf{S}$, i.e., $\exists Z \in \mathbf{S}$ s.t. $Z \in p$. We denote the set of frontiers between $X$ and $Y$ in $\mathcal{G}$ as $Fr_{\mathcal{G}}(X, Y)$.

**Proposition 4.5.** *Given $X \in \mathbf{X}$ and $Y$, and a subset $\mathbf{S} \in Fr_{\mathcal{G}}(X, Y)$, then $\nu(\mathbf{S} \cup \{X\}) = \nu(\mathbf{S})$.*

*Remark* 4.6. For any parent $X \in Pa_Y$, no subset $\mathbf{S} \subseteq \mathbf{X} \setminus \{X\}$ is a frontier between $X$ and $Y$.

**Theorem 4.7.** *Consider any subset $\mathbf{S} \subseteq \mathbf{X}$ and let us define $\mathbf{Z} := \{X \in \mathbf{S} \mid \mathbf{S}_{>_{\mathcal{G}} X} \in Fr_{\mathcal{G}}(X, Y)\}$, where $\mathbf{S}_{>_{\mathcal{G}} X} := \{Z \in \mathbf{S} \mid Z >_{\mathcal{G}} X\}$. Then $\nu(\mathbf{S}) = \nu(\mathbf{S} \setminus \mathbf{Z})$, and $\mathbf{S} \setminus \mathbf{Z}$ cannot be further reduced.*

Thanks to this theorem, we can significantly reduce execution time for do-SHAP by using a cache on these irreducible subsets, thereby avoiding the computation of any $\nu(\mathbf{S})$ term whose irreducible $\mathbf{S} \setminus \mathbf{Z}$ overlaps with a previously computed coalition. Additionally, we propose the Frontier-Reducibility Algorithm (FRA), described in Algorithm 1, to efficiently reduce any coalition $\mathbf{S} \subseteq \mathbf{X}$, encoded as $s := \sum_{V_k \in \mathbf{S}} 2^k$. Given three pre-computed maps and the `Frontier` map (populated as we execute the FRA procedure), we can employ the output set, uniquely encoded as an integer, to identify $\mathbf{S} \setminus \mathbf{Z}$, which will then be passed to $\nu$ and stored in the $\nu$-cache for subsequent evaluations.

Finally, note that this contribution is related to Luther et al. (2023) work, in which coalitions such that $\nu(\mathbf{S} \cup \{X\}) = \nu(\mathbf{S})$ were identified, but with $\nu$ defined for conditional SHAP. We move further by: 1) extending this idea to the do-SHAP causal setting, which requires the use of do-calculus to derive these properties; and 2) by describing and efficiently computing the irreducible set such that $\nu(\mathbf{S}) = \nu(\mathbf{S'})$, with $\mathbf{S'} \subseteq \mathbf{S}$. See Appendix D.2 for an in-depth explanation of the procedure and its preceding results.

### 4.3 DO-SHAPLEY EXPLANATIONS

So far, we have been talking about do-SHAP values *w.r.t.* a variable $Y \in \mathcal{V}$ in a certain SCM $\mathcal{M}$, but there are two use cases to consider in practice: either we want to explain a ML model that models $Y$ given some inputs $\mathbf{Z} \subseteq \mathcal{V} \setminus \{Y\}$ or we want to explain the original variable $Y$ directly.

If we want to explain a ML model $f(\mathbf{Z}) := \mathbb{E}[Y \mid \mathbf{Z}]$, we can replace $Y$ with $Y' := f(\mathbf{Z})$ ($E_Y$ will have no effect on $f_{Y'}$ since $f$ is deterministic) and then work on the projected SCM $\mathcal{M}[An(Y')]$ considering $Pa_Y = \mathbf{Z}$. Note that this subgraph may contain variables other than those in $\mathbf{Z}$, since any $X \in \mathbf{X} \setminus \mathbf{Z}$ may have an effect on some $Z \in \mathbf{Z}$, and are therefore ancestors of $Y$. With this SCM, we can apply estimand-agnostic procedures to estimate do-SHAP. We exemplify this case in the experiment in Section 5.1.

If, instead, we want to explain the target variable $Y$ directly, we simply employ do-SHAP on a proxy SCM, but note that for a particular $(\mathbf{x}, y) \sim \mathcal{P}(\mathbf{X}, Y)$, $\sum_{X \in \mathbf{X}} \phi_{\nu_{\mathbf{x}}}(X) = \mathbb{E}[Y \mid \widehat{\mathbf{x}}] - \mathbb{E}[Y] \neq y - \mathbb{E}[Y]$ (unless $Y$ is a constant distribution). There is a *gap* between the contribution of $\mathbf{X}$ ($\Delta_{\nu_{\mathbf{x}}}(\mathbf{X})$) and the actual value of $Y$, because our particular $\nu$, an interventional query, is essentially a population estimate, and as such aggregates for the whole distribution. In order to explain a particular outcome, we need some kind of *counterfactual* value function $\nu$; this is a promising avenue of research, but is left for future work, since it is beyond the scope of this paper. As an alternative approach, the following theorem proves that, under additional assumptions, we can explain this gap through $E_Y$'s SHAP attribution.

**Theorem 4.8.** *do-Shapley Value for the Noise.*
*Let us assume that $f_Y \in \mathcal{F}$ follows an additive noise model, i.e., $Y = f(Pa_Y) + E_Y$ for a certain function $f$. Consider the do-Shapley game involving variables $\mathbf{X}$ and $E_Y$. Then,*

---

**Algorithm 1** Frontier-Reducibility Algorithm (FRA)

---

**Require:** `ParentsY` $:= \sum_{V_k \in Pa_{\mathcal{G}}(Y)} 2^k$.

**Require:** $\forall k = 0..K - 1$, `Descendants`$[2^k] := \sum_{V_{k'} \in De_{\mathcal{G}}(V_k)} 2^{k'}$.

**Require:** $\forall k = 0..K - 1$, `Children`$[2^k] := \sum_{V_{k'} \in Ch_{\mathcal{G}}(V_k)} 2^{k'}$.

**Require:** `Frontier`, a map int $\to$ bool.

1: **procedure** IS-FRONTIER($s, x, y$, `Children`)
2:     $c \leftarrow x$                                                         ▷ Current nodes
3:     **while** $c \neq 0$ and $y$ & $c = 0$ **do**                    ▷ While $\mathbf{C} \neq \varnothing$ and $Y \notin \mathbf{C}$
4:         $s \leftarrow s \mid c$                                    ▷ Update visited nodes with the new nodes
5:         $c' \leftarrow c$
6:         **while** $c' > 0$ **do**                                  ▷ Iterate over the elements in $\mathbf{C}$
7:             $x \leftarrow 2^{\lfloor \log_2 c' \rfloor}$
8:             $c' \leftarrow c' - x$
9:             $c \leftarrow c \mid$ `Children`$[x]$
10:         **end while**
11:         $c \leftarrow c$ & $\neg s$                               ▷ Remove any previously visited nodes
12:     **end while**
13:     **return** c $= 0$
14: **end procedure**
15: **procedure** FRA($s$, `ParentsY`, `Descendants`, `Children`, `Frontier`)
16:     $p \leftarrow 0; z \leftarrow 0;$                                ▷ Initialize $\mathbf{P}, \mathbf{Z}$ (encoded)
17:     **while** $s > 0$ **do**
18:         $x \leftarrow 2^{\lfloor \log_2 s \rfloor}$                   ▷ Get the last element (encoded)
19:         $s \leftarrow s - x$
20:         **if** $x$ & `ParentsY` $= 0$ **then**                       ▷ Only if not a parent of $Y$
21:             $p' \leftarrow p$ & `Descendants`$[x]$                  ▷ Only check descendants
22:             $t \leftarrow p' + x$                             ▷ $p' + x$ uniquely defines $(p', x)$
23:             **if** $t \notin$ `Frontier` **then**
24:                 $v \leftarrow$ IS-FRONTIER($p', x, y$, `Children`)
25:                 `Frontier`$[t] \leftarrow v$
26:             **else**
27:                 $v \leftarrow$ `Frontier`$[t]$
28:             **end if**
29:             **if** $v$ **then**
30:                 $z \leftarrow z + x$
31:             **end if**
32:         **end if**
33:         $p \leftarrow p + x$
34:     **end while**
35:     **return** $p - z$                                       ▷ Return the encoded set $\mathbf{S} \setminus \mathbf{Z}$
36: **end procedure**

---

$\phi_{E_Y} = y - \mathbb{E}[Y \mid pa_Y]$ *and every other do-Shapley value $\phi_X$ for $X \in \mathbf{X}$ can be computed w.r.t. $\mathbf{X}$ only. Furthermore, $\sum_{X \in \mathbf{X}} \phi_X + \phi_{E_X} = y - \mathbb{E}[Y]$.*

Thanks to this theorem (proved in Appendix D.3), assuming an additive noise model for the target variable, we can explain inaccessible DGPs with attribution to the noise with no computational overhead. In practice, we can define a ML model $f'(pa_Y)$ for $\mathbb{E}[Y \mid pa_Y]$ and explain it instead, computing the do-SV for $E_Y$ afterwards with a simple subtraction, $\phi_{E_Y} = y - f'(pa_Y)$.

### 4.4 LIMITATIONS

We finish this section by discussing the limitations of our approach. The causal graph $\mathcal{G}$ must be known (otherwise, we would need domain experts and/or Causal Discovery algorithms (Spirtes & Zhang, 2016) to derive it), it must be a DAG, its distribution $P(\mathcal{V})$ must be strictly positive on its support, and the do-SV must be identifiable in $\mathcal{G}$. These limitations are shared with estimand-based

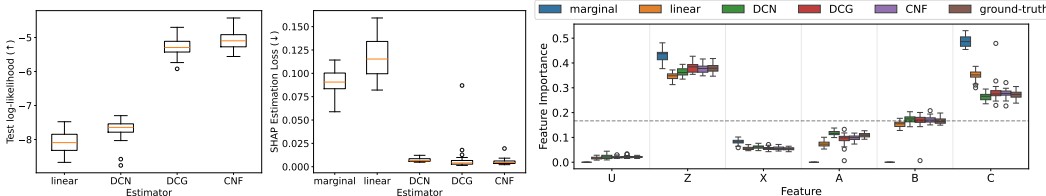

Figure 2: Markovian case. Box-plots computed over 30 realizations of the dataset. (a) Distribution adjustment score, log-likelihood (bigger is better). (b) SHAP estimation loss, $\mathcal{L}$ (lower is better). (c) Feature Importance (the closer to *ground-truth*, the better). Dashed horizontal line represents uniform importance ($\frac{1}{K}$). See Appendix E.1 for a bigger figure.

approaches, but, on the other hand, there is no doubly-robust general solution for SCMs yet, so this is a definite disadvantage *w.r.t.* estimand-based methods; nevertheless, their ad-hoc nature makes them impractical for do-SHAP, while our approach can adapt to arbitrary graphs. Finally, FRA requires that not all variables are parents of $Y$, since no coalition would be reducible otherwise. Fortunately, real-world DGPs rarely have all (proper) $Y$-ancestors as parents, and in the case of ML systems, defining all $\mathbf{X}$ as model inputs ($Pa_{Y'}$) is rarely advisable, since they may contain non-ancestors of $Y$ (leading to spurious correlations or anti-causal directions (Schölkopf et al., 2021)) or inputs $\mathbf{A} \subseteq \mathbf{X} \setminus Pa_Y$ that are blocked by $Pa_Y$, ($Y \perp\!\!\!\perp \mathbf{A} \mid Pa_Y$), in which case their inclusion could easily lead to overfitting and adversarial vulnerability. In fact, feature selection strategies should aim at discarding these cases.

## 5 Experiments

The *Fundamental Problem of Causal Inference* (Pearl, 2009) means that we can never observe causal effects on a single sample; when we observe an outcome (*factual*), we cannot go back in time to apply an intervention to obtain a different outcome (*counterfactual*), so we cannot measure the effect of that intervention. For this reason, our first experiment deals with synthetic data, for which we have access to the underlying DGP, to measure do-SHAP estimation error of several estimand-agnostic estimators. Secondly, we demonstrate the speedup resulting from the FRA algorithm, also on a synthetic dataset due to the difficulty in finding real datasets with known causal graphs and a progressively increasing number of features. Please refer to Appendix E for further details about these experiments. Finally, we showcase do-SHAP explanations on two real world datasets (a classification and a regression task). Given that the synthetic experiments validate our approach, while the real world experiments are mere examples of its applicability, we prioritize the former in terms of space and leave the latter for Appendix F due to space restrictions.

### 5.1 Estimation performance

We designed a synthetic SCM $\mathcal{M}_0$ with the graph in Figure 3 for two cases: assuming $U_{\{X,B\}}$ is observed (Markovian) or latent (semi-Markovian). We train a ML model $f(pa_X) \approx \mathbb{E}\left[Y \mid pa_Y\right]$, which will become the *accessible* DGP to explain. Consider a new SCM $\mathcal{M}$ based on $\mathcal{M}_0$ but with $Y$ replaced by $Y' := f(Pa_Y)$; let $\mathbf{X} := \mathcal{V} \setminus \{Y'\}$. Note that both cases are identifiable.

We replicate the experiment 30 times with different seeds. Let $\mathcal{D}$ be a dataset generated from $\mathcal{M}$ with $N = 1000$ i.i.d. samples. Since we have access to the DGP, we can estimate each query $\nu(\mathbf{S})$ by Monte Carlo with $M$ i.i.d. samples from the intervened DGP, passing them through $f$ and averaging the outputs; we will use the do-Shapley values $\Phi := (\phi_X^{(i)})_{i \in [N], X \in \mathbf{X}}$ computed from these estimations as *ground truth*. We will train several kinds of SCM (with $Y'$ replacing $Y$) to learn the distribution $\mathcal{P}(\mathcal{V})$, use them to estimate the do-Shapley values $\tilde{\Phi} = (\tilde{\phi}_X^{(i)})_{i \in [N], X \in \mathbf{X}}$, and compute their **SHAP** estimation **loss** $\mathcal{L}_2(\Phi, \tilde{\Phi}) := \frac{1}{N|\mathbf{X}|} \sum_{i=1}^{N} \sum_{k=1}^{|\mathbf{X}|} (\phi_{X_k}^{(i)} - \tilde{\phi}_{X_k}^{(i)})^2$. We will also compare against a marginal-SHAP estimator (which should result in different values). We compute the average test log-likelihood (**loglk**) for each model as a way to measure distribution

adjustment. Finally, for all $X \in \mathbf{X}$, we compute its Feature Importance (**FI**), defined as $FI_X :=$
$\frac{1}{N} \sum_{i \in [N]} \frac{|\phi_X^{(i)}|}{\sum_{X' \in \mathbf{x}} |\phi_{X'}^{(i)}|}$.

We will test do-SHAP with several SCM architectures to compare among them[6]; we justify our choices in Appendix E.1.1, along with further implementation details, due to space restrictions. These methods are: 1) a **linear** SCM with Normal distributions for each variable, used as a baseline; 2) the Distributional Causal Node (Parafita & Vitrià, 2019) (**DCN**), where every node is modeled after a specific distribution; and 3) Deep Causal Graph (Parafita & Vitrià, 2022) (**DCG**) powered with Normalizing Flows. Additionally, in order to test the alternative approach of modeling SCMs not node-wise, but the graph as a whole, we opt for Causal Normalizing Flows (**CNF**) (Javaloy et al., 2024).

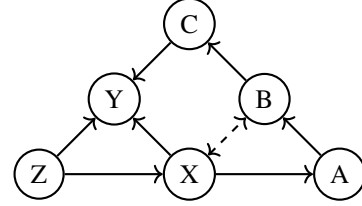

Figure 3: Synthetic semi-Markovian graph. The Markovian graph results from $U_{\{X,B\}}$ (see Footnote 2) being observed.

See Figure 2 for the Markovian case. As expected, distribution adjustment (loglk) correlates with SHAP estimation performance; as our SCMs better model $\mathcal{P}(\mathcal{V})$, they better estimate $\nu(\mathbf{S})$, resulting in better do-SHAP estimations. Linear-SCM cannot adjust properly to the dataset's distribution, and so its do-SHAP performance suffers; DCN comes remarkably close to the best two models, probably because of the synthetic nature of the data; DCGs and CNFs exhibit similar performance, with DCGs having more variance, possibly due to CNFs modeling all variables at once. Finally, marginal SHAP significantly differs from the do-SHAP ground truth, showing that, evidently, do-SHAP and marginal-SHAP measure different kinds of importance. FI comparisons *w.r.t.* ground truth values are also aligned with the previous conclusions. As for the semi-Markovian experiment, we found equivalent conclusions even without measuring the latent confounder, with DCGs displaying the best estimation performance and FI values in agreement with the ones in the Markovian case. We leave this experiment to Appendix E.1.2.

In the following, we will employ DCGs instead of CNFs because, while CNFs seem to be more stable variance-wise, DCGs admit latent confounders and are orders of magnitude faster than CNFs.

## 5.2 FRONTIER-REDUCIBILITY ALGORITHM

We now test the speed-up resulting from the FRA; do-SHAP estimation performance is not tested here since it is already covered in the previous experiment. Let us consider $\mathcal{G}_{K,p}$, the class of graphs $\mathcal{G}$ with $K+1$ nodes, defined in topological order, $\mathbf{X} := (V_0, \ldots, V_{K-1}), Y := V_K$, where $p \in (0,1)$ is a parameter such that, for any possible edge $V_i \to V_j, 0 \leq i < j \leq K$, the probability of this edge appearing in $\mathcal{G}$ is $p$, and such that they fulfill two conditions: 1) $An(Y) = \mathbf{X} \cup \{Y\}$ and 2) $Pa_Y \subsetneq \mathbf{X}$ (otherwise, FRA will trivially have no effect and should be skipped). It is trivial to sample a graph from this distribution using rejection sampling to ensure that both conditions are fulfilled.

Figure 4 summarizes our experiments. We will compute the error bars for the mean of several metrics at 2-sigma over 30 graphs per configuration. Let $K \in \{5, \ldots, 20\}$ and $p \in \{0.1, \ldots, 0.9\}$. Figure 4 (a) shows the average ratio of coalitions out of the $2^K$ possible coalitions that need to be evaluated with $\nu$ after reduction by FRA. Note that, while these ratios depend on the actual topology of the graph, each $p$-curve approaches the region of $p$, which in the case of $p = 0.1$ leads to a 90% reduction in $\nu$ computations. Figure 4 (b) shows the average execution time of FRA per coalition. Despite the exponentially-larger number of directed paths in the graph (in the worst case, a complete graph, $2^K - 1$ directed paths from $X \in \mathbf{X}$ to $Y$), the computation of FRA appears to grow linearly with $K$, due to the fact that it scales with the size of the coalition $\mathbf{S}$ to be evaluated ($|S| < K$) and the depth of the graph (at most $K$). For $K \leq 20$, the error bars do not exceed $3\mu s$.

Finally, we evaluate FRA with an ablation test in Figure 4 (c). We design synthetic DGPs for random $\mathcal{G} \in \mathcal{G}_{K,p}$ with $\forall X \in \mathcal{V}, f_X(pa_X, \varepsilon_X) := \text{mean}(pa_X) + \varepsilon_X, \varepsilon_X \sim \mathcal{N}(0,1)$. We choose a linear SCM for its fast execution; real world SCMs, with far more complex architectures, will

---

[6]None of these methods are external baselines, since do-SHAP has not been tested with estimand-agnostic approaches yet, nor has seen much use so far because of the ad hoc nature of estimand-based approaches.

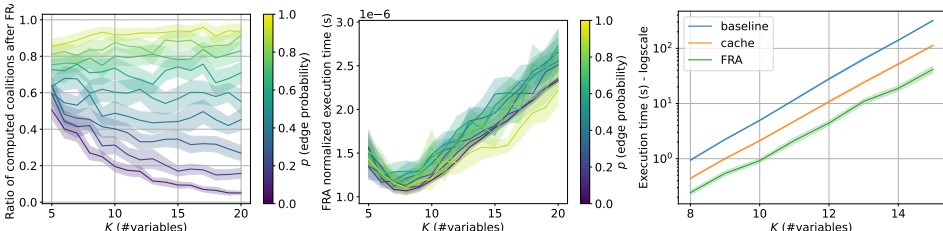

Figure 4: FRA experiments. (a) Ratio of computed coalitions after FRA. (b) FRA execution time per coalition. (c) do-SHAP execution time (logarithmic scale) without cache (*baseline*), with cache (*cache*) and with an FRA cache (*FRA*). Error bars at 2-sigma over 30 replications.

require even longer to execute, and FRA will therefore have an even stronger impact. We evaluate do-SHAP with a linear DCG and using the approximate method with $N$ permutations such that the ratio of processed coalitions after $N$ permutations is bigger than $0.5$, computed with Equation (6) in Appendix B; we set $K \geq 8$ so that $N \geq 30$. Note that this choice for $N$ results in an exponential time-growth *w.r.t.* $K$. We restrict this experiment to $8 \leq K \leq 15$ and $p = 0.25$. For the ablation test, we run three alternatives: compute every coalition $\mathbf{S}$ (**baseline**), employ a **cache**, and employ an FRA cache (**fra**). We plot mean execution time for each alternative, with a consistent pattern: FRA is an order of magnitude faster than the baseline and twice faster than the cache.

Please refer to Appendix E.2 for further tests on FRA and our experimental setup. There we show that FRA's execution time is negligible *w.r.t.* the computation of $\nu(\mathbf{S})$, even on linear SCMs. This difference can only increase with more complex SCM architectures; therefore, for virtually no cost, FRA skips computing $\nu(\mathbf{S})$ up to a significant factor, resulting in a marked speedup for do-SHAP.

## 6 CONCLUSION

In this work, we have introduced a practical and scalable method to estimate do-SVs using the estimand-agnostic approach, with which we can estimate any identifiable query—in particular, do-SVs—using general procedures agnostic to the query's estimand. This flexibility is essential to make these techniques accessible to practitioners, who may not necessarily be experts in Causal Inference. We have tested our approach on multiple SCM architectures, showcasing the relationship between distribution modeling and do-SHAP estimation performance, which paves the way for future research. We have demonstrated several do-SHAP properties along with the proposal of the Frontier-Reducibility Algorithm to speed up do-SHAP significantly. Finally, we have applied our method on two real-world datasets (see Appendix F) showcasing the applicability of these techniques to obtain reliable explanations, either from a ML model or an inaccessible DGP.

Further work could propose new SCM architectures to better model the data distribution, along with more efficient estimators (ideally with doubly-robust guarantees) for the causal queries underlying do-SHAP. A general graphical criterion for do-SV identifiability is also a worthwhile new direction. Finally, do-SVs are based on interventional queries, but these are inherently population-based measures, and therefore not really appropriate for individual, local explanations; alternative definitions of the value function could try to offer causal local explanations.

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

# A  SHAPLEY VALUE AXIOMS

The Shapley value $\phi = \{\phi_X\}_{X \in \mathbf{X}}$ is the unique attribution measure that fulfills a number of desirable properties:

- **Efficiency**: $\sum_{X \in \mathbf{X}} \phi_X = \nu(\mathbf{X}) - \nu(\varnothing) = \Delta(\mathbf{X})$; the sum of Shapley values adds up to the total contribution of $\mathbf{X}$.

- **Missingness**: if $\forall \mathbf{S} \subseteq \mathbf{X} \setminus \{X\}$, $\nu(\mathbf{S} \cup \{X\}) = \nu(\mathbf{S})$, then $\phi_X = 0$; players with no contribution to any coalition have Shapley value 0.

- **Symmetry**: if $\forall \mathbf{S} \subseteq \mathbf{X} \setminus \{X, Y\}, \nu(\mathbf{S} \cup \{X\}) = \nu(\mathbf{S} \cup \{Y\})$ then $\phi_X = \phi_Y$; players with identical contribution to any coalition have identical Shapley values.

# B  CACHE IMPACT ON THE APPROXIMATION ALGORITHM

Consider the approximation method (see Section 3.4), where we sample permutations of $K$ elements uniformly with replacement, $\pi \sim \mathcal{U}(\Pi([K]))$, so as to approximate the Shapley value with a Monte Carlo estimator. In this section, we want to evaluate how much we can accelerate the computation of new permutations as we fill a cache with the values of previously computed coalitions. When we use a cache, once we compute a coalition for the first time, we save its result in it (assuming no cache limit) and further computations of this coalition will incur in negligible computation time (simply a cache access), therefore speeding up the computation of new permutations. We want to measure exactly how much we can speed up the process.

Let us define some notation. Given $\pi \in \Pi([K])$, let us denote by $\mathcal{C}(\pi)$ the set of $K + 1$ coalitions $\mathbf{S} \in \mathbb{P}([K])$ defined by taking the first $s$ elements of $\pi$, $s = 0..K$ (e.g., for $\pi = (3, 1, 2)$, $\mathcal{C}(\pi) = \{\varnothing, (3), (3, 1), (3, 1, 2)\}$). Then, for an arbitrary $\mathbf{S} \in \mathbb{P}([K])$ and a permutation $\pi \sim \mathcal{U}(\Pi([K]))$:

$$P(\mathbf{S} \in \mathcal{C}(\pi)) = \frac{|\mathbf{S}|!(K - |\mathbf{S}|)!}{K!} = \binom{K}{|\mathbf{S}|}^{-1} \tag{4}$$

since $\mathbf{S}$ must appear at the beginning of $\pi$ in an arbitrary order, so there is $|\mathbf{S}|!$ possibilities, with the remaining $(K - |\mathbf{S}|)$ elements in an arbitrary order, so $(K - |\mathbf{S}|)!$, out of the total $K!$ possible permutations. Since we are taking $N$ i. i. d. permutations $(\pi^{(n)})_{n \in [N]}$, it follows that

$$P(\forall n \in [N], \mathbf{S} \notin \mathcal{C}(\pi^{(n)})) = \left(1 - \binom{K}{|\mathbf{S}|}^{-1}\right)^N, \tag{5}$$

which is the probability of an arbitrary coalition $\mathbf{S}$ not belonging to any of the $N$ previously sampled permutations, and therefore, it still needs to be computed when it appears in a future permutation. In particular, note that we do not need to know the elements of $\mathbf{S}$, only its cardinality $|\mathbf{S}|$, which we will denote by $s := |\mathbf{S}|$. Given the set of $K + 1$ coalitions $\mathcal{C}(\pi^{(N)})$ in permutation $\pi^{(N)}$, we can now compute the expected ratio of its coalitions not found in any of the previous permutations

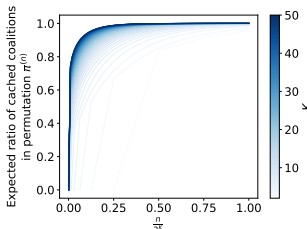 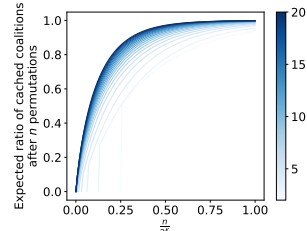

Figure 5: Cache evolution plots. a) Ratio of coalitions in $\pi^{(n)}$ already cached. b) Ratio of total coalitions already cached after $n$ permutations. Both x-axis represent the number of permutations $n$ divided by $2^K$, so as to compare between different values of $K$.

(therefore not cached); in other words, the expected ratio of computations we need to perform at the $N$-th permutation, $N > 1$, is:

$$\frac{1}{K+1} \sum_{\mathbf{S} \in \mathcal{C}(\pi^{(N)})} P(\forall n \in [N-1], \mathbf{S} \notin \mathcal{C}(\pi^{(n)})) = \frac{1}{K+1} \sum_{s=0}^{K} \left(1 - \binom{K}{s}^{-1}\right)^{N-1}. \tag{6}$$

For $N = 1$, the ratio is trivially 1. Also that for $s = 0$ ($\mathbf{S} = \varnothing$) and $s = K$ ($\mathbf{S} = [K]$), the term $\left(1 - \binom{K}{s}^{-1}\right)$ becomes 0 (it is impossible not to have seen them in a previous permutation, since they are in every permutation), so we omit these cases in the following sums.

Finally, the expected ratio of cached coalitions (out of the total number of coalitions $2^K$) after $N \geq 1$ permutations is:

$$\frac{1}{2^K} \sum_{n=1}^{N} \sum_{\mathbf{S} \in \mathcal{C}(\pi^{(n)})} P(\forall n' \in [n-1], \mathbf{S} \notin \mathcal{C}(\pi^{(n')})) = \frac{K+1}{2^K} + \frac{1}{2^K} \sum_{n=2}^{N} \sum_{s=1}^{K-1} \left(1 - \binom{K}{s}^{-1}\right)^{n-1}$$

$$= \frac{K+1}{2^K} + \frac{1}{2^K} \sum_{s=1}^{K-1} \binom{K}{s}\left(1 - \binom{K}{s}^{-1}\right)\left(1 - \left(1 - \binom{K}{s}^{-1}\right)^{N-1}\right)$$

$$= \frac{K+1}{2^K} + \frac{1}{2^K} \sum_{s=1}^{K-1} \left(\binom{K}{s} - 1\right) - \frac{1}{2^K} \sum_{s=1}^{K-1} \binom{K}{s}\left(1 - \binom{K}{s}^{-1}\right)^{N}$$

$$= 1 - \frac{1}{2^K} \sum_{s=0}^{K} \binom{K}{s}\left(1 - \binom{K}{s}^{-1}\right)^{N}, \tag{7}$$

where we first split the sum over $n$ for $n = 1$ and $n > 1$, and then swap the sums and apply, for $x := 1 - \binom{K}{s}^{-1}$, the equality $\sum_{n=1}^{N} x^n = x \frac{1-x^N}{1-x}$ for $x \in (0,1)$ (which is the case when $s \neq 0, K$), and noting that $\frac{1}{1-x} = \binom{K}{s}$. We then split the sum in two terms, with the first half adding up to 1 with $\frac{K+1}{2^K}$. The rest of the transformation is trivial.

We now plot Equations (6) and (7) in Figure 5 (a) and (b), respectively, for several values of $K$ (represented by color opacity). The x-axis in both cases is $\frac{n}{2^K}$, so as to show how each curve progresses as $n \to 2^K$, where we will have encountered $(K+1)2^K$ coalitions. We can see: a) that the likelihood of encountering previously-computed coalitions is very high early in the process, which means that the computations required per permutation speed up significantly in the early stages; b) the fraction of the total number of coalitions requires many more permutations to approach 100%. These plots are merely illustrative; we encourage researchers to make use of the derived equations to adjust for the appropriate number of permutations in terms of computation time budget.

## C  CAUSAL INFERENCE CONCEPTS

We include here some additional notation and concepts for Causal Inference, necessary for the proofs in Appendix D.

NOTATION

Given r.v.s $X \neq Y$ and a disjoint set of r.v.s $\mathbf{Z}$ (possibly empty), we denote that $X$ is independent of $Y$ conditioned on $\mathbf{Z}$ in a distribution $\mathcal{P}$ by $(X \perp\!\!\!\perp Y \mid \mathbf{Z})_{\mathcal{P}}$. Given disjoint sets of r.v.s $\mathbf{X}, \mathbf{Y}, \mathbf{Z}$, we say that $\mathbf{X}$ is independent of $\mathbf{Y}$ given $\mathbf{Z}$ in a distribution $\mathcal{P}$, denoted by $(\mathbf{X} \perp\!\!\!\perp \mathbf{Y} \mid \mathbf{Z})_{\mathcal{P}}$, if and only if $\forall X \in \mathbf{X}, \forall Y \in \mathbf{Y}, (X \perp\!\!\!\perp Y \mid \mathbf{Z})_{\mathcal{P}}$. $\mathcal{P}$ can be omitted unless it leads to ambiguity.

Given $\mathbf{X}, \mathbf{Y} \subseteq \mathbf{V}$, let $\mathcal{G}_{\overline{\mathbf{X}}\underline{\mathbf{Y}}}$ denote the graph $\mathcal{G}$ modified such that all edges pointing towards nodes in $\mathbf{X}$ are removed (overline) and all edges starting from nodes in $\mathbf{Y}$ are removed (underline). We may incur in abuse of notation (e.g., $\mathcal{G}_{\overline{\mathbf{X}Y}} := \mathcal{G}_{\overline{\mathbf{X} \cup \{Y\}}}$) unless it leads to ambiguity.

## C.1 $d$-SEPARABILITY AND $do$-CALCULUS

In the following, we will define the concept of $d$-separability, its connection to independence, and the three rules of $do$-**calculus**. Please refer to (Pearl, 2009) for more details.

**Definition C.1.** $d$-**separability**.
Given a DAG $\mathcal{G} = (\mathbf{V}, \mathbf{E})$, a path $p$ is $d$-*separated* (blocked) by a set $\mathbf{Z} \subseteq \mathbf{V}$ (possibly empty) if and only if either is true:

1. $p$ contains a *chain* $A \to B \to C$ or a *fork* $A \leftarrow B \to C$ such that $B$ is in $\mathbf{Z}$.

2. $p$ contains a *collider* $A \to B \leftarrow C$ such that no descendant of $B$ (including $B$) is in $\mathbf{Z}$.

Given disjoint sets $\mathbf{X}, \mathbf{Y}, \mathbf{Z} \subseteq \mathbf{V}$, we say that $\mathbf{Z}$ *d-separates $X$ from $Y$* in $\mathcal{G}$ if $\mathbf{Z}$ $d$-separates every path $p$ from a node $X \in \mathbf{X}$ to a node $Y \in \mathbf{Y}$. We denote this by $(\mathbf{X} \perp\!\!\!\perp \mathbf{Y} \mid \mathbf{Z})_{\mathcal{G}}$.

**Definition C.2. Markov Compatibility**.
We say that a distribution $\mathcal{P}(\mathcal{V})$ on a set of variables $\mathcal{V} = (V_1, \cdots, V_K)$ is (Markov) *compatible* with a DAG $\mathcal{G}$ with $\mathcal{V}$ as vertices in $\mathcal{G}$ if $P(\mathcal{V}) = \prod_{k \in [K]} \mathcal{P}(V_k \mid Pa_{\mathcal{G}}(V_k))$.

**Theorem C.3.** *Independence and $d$-separability*.
*Given an SCM $\mathcal{M} = (\mathcal{V}, \mathcal{W}, \mathcal{P}, \mathcal{F})$ compatible with a DAG $\mathcal{G}_{\mathcal{M}}$ and disjoint sets $X, Y, Z \subseteq \mathcal{V}$, if $(X \perp\!\!\!\perp Y \mid Z)_{\mathcal{G}_{\mathcal{M}}}$ then $(X \perp\!\!\!\perp Y \mid Z)_{\mathcal{P}}$. Conversely, if $(X \not\perp\!\!\!\perp Y \mid Z)_{\mathcal{G}_{\mathcal{M}}}$, there exists at least one distribution $\mathcal{P}'$ compatible with $\mathcal{G}_{\mathcal{M}}$ (in fact, almost all) such that $(X \not\perp\!\!\!\perp Y \mid Z)_{\mathcal{P}'}$.*

*Remark* C.4. The second statement comes from the fact that precise parameter choices $\theta$ of distributions $\mathcal{P}_{\Theta}$ might result in independence in an otherwise unblocked path in $\mathcal{G}$. Fortunately, such specific tuning of $\Theta$ rarely occurs in practice.

*Remark* C.5. If we need to determine independence relationships $(\mathbf{X} \perp\!\!\!\perp \mathbf{Y} \mid \mathbf{Z})_{\mathcal{P}}$ ($\mathbf{Z}$ possibly empty), we simply verify that all paths connecting $\mathbf{X}$ and $\mathbf{Y}$ are blocked by $\mathbf{Z}$, using $d$-separability.

Next, we introduce the three rules of $do$-calculus, with which we can transform causal queries step by step, until we reach the desired estimand.

**Theorem C.6.** *Rules of $do$-calculus*.
*Given an SCM $\mathcal{M} = (\mathcal{V}, \mathcal{W}, \mathcal{P}, \mathcal{F})$ compatible with a DAG $\mathcal{G}_{\mathcal{M}}$, for any disjoint sets $X, Y, Z, W, \subseteq \mathcal{V}$ ($X$ and $W$ possibly empty):*

1. *Insertion/deletion of observations (R1):*
   $P_x(Y \mid Z, W) = P_x(Y \mid W)$    *if $(Y \perp\!\!\!\perp Z \mid X, W)_{\mathcal{G}_{\overline{X}}}$.*

2. *Exchange of interventions/observations (R2):*
   $P_{x,z}(Y \mid W) = P_x(Y \mid z, W)$    *if $(Y \perp\!\!\!\perp Z \mid X, W)_{\mathcal{G}_{\overline{X}\underline{Z}}}$.*

3. *Insertion/deletion of interventions (R3):*
   $P_{x,z}(Y \mid W) = P_x(Y \mid W)$    *if $(Y \perp\!\!\!\perp Z \mid X, W)_{\mathcal{G}_{\overline{X}\overline{Z(W)}}}$,*
   *where $Z(W) := Z \setminus An_{\mathcal{G}_{\overline{X}}}(W)$, the set of nodes in $Z$ that are not ancestors of $W$ (including $W$) in the graph $\mathcal{G}_{\overline{X}}$.*

## C.2 Projected Structural Causal Models

**Definition C.7. Divergent Path**.
A *divergent path* between $X$ and $Y$ consists of two directed paths, from $W$ to $X$ and from $W'$ to $Y$, such that $W = W'$ or $W \leftrightarrow W'$.

**Definition C.8. Projected SCM**.
Given an SCM $\mathcal{M} = (\mathcal{V}, \mathcal{W}, \mathcal{P}, \mathcal{F})$ compatible with a DAG $\mathcal{G}_{\mathcal{M}}$ and a subset $\mathcal{V}' \subseteq \mathcal{V}$, we define the *projected causal DAG* $\mathcal{G}[\mathcal{V}']$ defined on vertices $\mathcal{V}'$ and $\mathcal{W}' := \mathcal{E}' \cup \mathcal{W}'$, with $\mathcal{E}' := \{E_X \in \mathcal{E} \mid X \in \mathcal{V}'\}$ and $\mathcal{U}'$ as defined next, such that:

- $\forall V_k, V_l \in \mathcal{V}'$, there is a directed edge $V_k \to V_l$ if there exists a directed path from $V_k$ to $V_l$ in $\mathcal{G}_{\mathcal{M}}$ where every internal node in the path is not in $\mathcal{V}'$.

- $\forall V_k, V_l \in \mathcal{V}'$, there is a bidirected edge $V_k \leftrightarrow V_l$ (connected by a latent confounder $U_{\{k,l\}} \in \mathcal{U}'$) if there exists a *divergent* path in $\mathcal{G}$ between them such that every internal node is not in $\mathcal{V}'$.

We define the *projected SCM* $\mathcal{M}[\mathcal{V}']$ by restricting its graph to $\mathcal{G}_{\mathcal{M}}[\mathcal{V}']$, with distribution $\mathcal{P}_{\mathcal{M}[\mathcal{V}']}(\mathcal{V}') = \mathcal{P}_{\mathcal{M}}(\mathcal{V}')$.

*Remark C.9.* The projected SCM respects all conditional independence relationships and the rules of $do$-calculus in the original graph. (Lee & Bareinboim, 2020).

# D Proofs

In this section, we will prove the results in the main paper and discuss the Frontier-Reducibility Algorithm.

## D.1 Non-ancestors

**Lemma D.1.** *Given a DAG $\mathcal{G} = (\boldsymbol{V}, \boldsymbol{E})$ and disjoint subsets of vertices $\boldsymbol{X}, \boldsymbol{Y} \subseteq \boldsymbol{V}$ (possibly empty), if there is a path $p$ in $\mathcal{G}_{\overline{\boldsymbol{X}}\underline{\boldsymbol{Y}}}$, then $p$ is a path in $\mathcal{G}$.*

*Proof.* $\mathcal{G}_{\overline{\boldsymbol{X}}\underline{\boldsymbol{Y}}}$'s edges are a subset of $\mathcal{G}$'s edges, since $\mathcal{G}_{\overline{\boldsymbol{X}}\underline{\boldsymbol{Y}}}$ only removes edges either ending in $\mathbf{X}$ or starting from $\mathbf{Y}$. Adding those edges back in $\mathcal{G}$ cannot remove any edge from the path; hence, $p$ is a path in $\mathcal{G}$. $\square$

**Proposition D.2.** *Non-Ancestors do not Contribute*.
*Let $\mathcal{M}$ be an SCM $\mathcal{M} = (\mathcal{V}, \mathcal{W}, \mathcal{P}, \mathcal{F})$, $Y$ the target r.v., $\boldsymbol{X}$ a subset $\boldsymbol{X} \subseteq \mathcal{V} \setminus \{Y\}$ and $\boldsymbol{x}$ a realization $\boldsymbol{x} \sim \mathcal{P}(\boldsymbol{X})$. For any $X \in \boldsymbol{X}$, if $X$ is not an ancestor of $Y$, then $\phi_{\nu_x}(X) = 0$.*

*Proof.* We will prove that $\forall X \notin An_{\mathcal{G}}(Y), \forall \mathbf{S} \subseteq \mathbf{X} \setminus \{X\}, \nu(\mathbf{S} \cup \{X\}) = \mathbb{E}\left[Y \mid \widehat{\mathbf{s}}, \widehat{x}\right] = \mathbb{E}\left[Y \mid \widehat{\mathbf{s}}\right] = \nu(\mathbf{S})$. If that is the case, then

$$\phi_X = \sum_{\mathbf{S} \subseteq \mathbf{X} \setminus \{X\}} \frac{1}{K} \binom{K-1}{|\mathbf{S}|}^{-1} (\nu(\mathbf{S} \cup \{X\}) - \nu(\mathbf{S})) = 0.$$

Note that $\mathbb{E}\left[Y \mid \widehat{\mathbf{s}}, \widehat{x}\right] = \mathbb{E}\left[Y \mid \widehat{\mathbf{s}}\right]$ if $P_{\mathbf{s},x}(Y) = P_{\mathbf{s}}(Y)$, which is implied by R3 if $(Y \perp\!\!\!\perp X \mid \mathbf{S})_{\mathcal{G}_{\overline{\mathbf{SX}}}}$.

Let us prove this independence by contradiction: assume there is a path $p$ connecting $X$ and $Y$ unblocked conditioned on $\mathbf{S}$ in $\mathcal{G}_{\overline{\mathbf{SX}}}$. The path cannot start with $X \leftarrow \cdots$ since all edges pointing towards $X$ are removed in $\mathcal{G}_{\overline{\mathbf{SX}}}$, so $p = X \to \cdots \overset{?}{-} Y$. Since the path is unblocked, if there were any left arrows ($\leftarrow$) in the path, the resulting collider $\cdots \to B \leftarrow \cdots$ must necessarily fulfill $De_{\mathcal{G}_{\overline{\mathbf{SX}}}}(B) \in \mathbf{S}$ to unblock the path. There are two cases: 1) if $B \in \mathbf{S}$, then there is an edge $B \leftarrow \cdots$ for a node $B \in \mathbf{S}$, which cannot be true in $\mathcal{G}_{\overline{\mathbf{SX}}}$; 2) if $B \in An_{\mathcal{G}_{\overline{\mathbf{SX}}}}(\mathbf{S}) \setminus \mathbf{S}$, then there is a directed path from $B$ to a node in $\mathbf{S}$, which again cannot happen in $\mathcal{G}_{\overline{\mathbf{SX}}}$ because we have removed all edges pointing towards $\mathbf{S}$. Therefore, the path must necessarily not contain any left arrows, which means that $p$ is a directed path from $X$ to $Y$ in $\mathcal{G}_{\overline{\mathbf{SX}}}$, which must also be a directed path in $\mathcal{G}$ due to Lemma D.1; therefore $X \in An_{\mathcal{G}}(Y)$, contradicting the initial assumption. No unblocked path can exist, which proves $(Y \perp\!\!\!\perp X \mid \mathbf{S})_{\mathcal{G}_{\overline{\mathbf{SX}}}}$ and the theorem in turn. $\square$

## D.2 Frontier-Reducibility Algorithm

We begin by defining the concept of *frontier* and proving several properties related to it, necessary for the definition of the Frontier-Reducibility Algorithm (FRA), introduced next. We finish with an alternative formulation of the FRA algorithm with integers for faster execution time and lesser memory usage.

### D.2.1 Frontiers and properties

In the following, consider an SCM $\mathcal{M} = (\mathcal{V}, \mathcal{W}, \mathcal{P}, \mathcal{F})$ with associated DAG $\mathcal{G} = (V, E)$, where $\mathcal{V} = (V_0, \ldots, V_K)$ is sorted in an arbitrary topological order of the graph. Let $\mathbf{X} := \{V_0, \ldots, V_{K-1}\}$, $Y := V_K$, and assume that $\mathbf{X} \subseteq An(Y)$. Note that there may be latent confounders ($\mathcal{U} \neq \varnothing$).

**Definition D.3.** Given any node $X \in \mathbf{X}$, a subset $\mathbf{S} \subseteq \mathbf{X}$ is a frontier between $X$ and $Y$ if $X \notin \mathbf{S}$ and all directed paths $p = (X, \ldots, Y)$ from $X$ to $Y$ are blocked by $\mathbf{S}$, i.e., $\exists Z \in \mathbf{S}$ s.t. $Z \in p$. We denote the set of frontiers between $X$ and $Y$ in $\mathcal{G}$ as $Fr_{\mathcal{G}}(X, Y)$.

**Proposition D.4.** *Given nodes $X \in \mathbf{X}$ and $Y$, and a subset $\mathbf{S} \in Fr_{\mathcal{G}}(X, Y)$, frontier from $X$ to $Y$, then $\nu(\mathbf{S} \cup \{X\}) = \nu(\mathbf{S})$.*

*Proof.* We will apply R3 by proving that $(Y \perp\!\!\!\perp X \mid \mathbf{S})_{\mathcal{G}_{\overline{\mathbf{S}X}}}$, in which case

$$\nu(\mathbf{S} \cup \{X\}) = \mathbb{E}\left[Y \mid \widehat{\mathbf{s}}, \widehat{x}\right] = \mathbb{E}\left[Y \mid \widehat{\mathbf{s}}\right] = \nu(\mathbf{S}).$$

Note that in $\mathcal{G}_{\overline{\mathbf{S}X}}$, all paths from $X$ to $Y$ are front-door paths. Consider any such $p = X \to \cdots \overset{?}{-} Y$. If the path is fully directed, since $\mathbf{S}$ is a frontier, $\exists Z \in \mathbf{S}$ s.t. $Z$ is in the path, thereby blocking it. If it is not directed, there exists a collider $\cdots \to Z \leftarrow \cdots$, which also blocks the path: $Z \notin An(\mathbf{S})$, since $\mathbf{S}$ has no ancestors other than itself in $\mathcal{G}_{\overline{\mathbf{S}X}}$ and $Z \notin \mathbf{S}$ because $\cdots \to Z$ is in $p$. Therefore, any path $p$ between $X$ and $Y$ must be blocked by $\mathbf{S}$ in $\mathcal{G}_{\overline{\mathbf{S}X}}$, which proves R3. □

*Remark* D.5. For any parent $X \in Pa_Y$, no subset $\mathbf{S} \subseteq \mathbf{X} \setminus \{X\}$ is a frontier between $X$ and $Y$.

**Proposition D.6.** *Given nodes $X \in \mathbf{X}$ and $Y$, and a frontier $\mathbf{S} \in Fr_{\mathcal{G}}(X, Y)$,*

  *1. $\forall \mathbf{S}' \subseteq \mathbf{X} \setminus \{X\}, \mathbf{S}' \supseteq \mathbf{S}$, then $\mathbf{S}' \in Fr_{\mathcal{G}}(X, Y)$.*

  *2. $\mathbf{S} \cap De(X) \in Fr_{\mathcal{G}}(X, Y)$.*

*Proof.*

  1. Since $\mathbf{S} \in Fr_{\mathcal{G}}(X, Y)$, any directed path $p$ between $X$ and $Y$ is blocked by $\mathbf{S}$; being $\mathbf{S}'$ a superset of $\mathbf{S}$, it must also block all such paths.

  2. Any non-descendant of $X$ cannot appear in a directed path from $X$ to $Y$, which means that it is superfluous in the frontier set. As such, $\mathbf{S} \cap De(X) \in Fr_{\mathcal{G}}(X, Y)$.

□

**Corollary D.7.** *Given $X \in \mathbf{X}$ and $\mathbf{S} \subseteq \mathbf{X} \setminus \{X\}$, let $\mathbf{S}_{>_{\mathcal{G}}X} := \{Z \in \mathbf{S} \mid Z >_{\mathcal{G}} X\}$.*

$$\mathbf{S} \in Fr_{\mathcal{G}}(X, Y) \Leftrightarrow \mathbf{S}_{>_{\mathcal{G}}X} \in Fr_{\mathcal{G}}(X, Y) \Leftrightarrow \mathbf{S} \cap De_{\mathcal{G}}(X) \in Fr_{\mathcal{G}}(X, Y). \tag{8}$$

*Proof.* If $\mathbf{S} \in Fr_{\mathcal{G}}(X, Y)$, $\mathbf{S} \cap De(X) \in Fr_{\mathcal{G}}(X, Y)$, and $\mathbf{S}_{>_{\mathcal{G}}X} \supseteq \mathbf{S} \cap De(X)$, since any $Z \in \mathbf{S} \cap De(X)$ fulfills $Z >_{\mathcal{G}} X$, which proves that $\mathbf{S}_{>_{\mathcal{G}}X} \in Fr_{\mathcal{G}}(X, Y)$. On the other hand, if $\mathbf{S}_{>_{\mathcal{G}}X} \in Fr_{\mathcal{G}}(X, Y)$, $\mathbf{S} \in Fr_{\mathcal{G}}(X, Y)$, since $\mathbf{S} \supseteq \mathbf{S}_{>_{\mathcal{G}}X}$. The remaining iff is trivial given Proposition D.6. □

**Definition D.8.** A set $\mathbf{S} \subseteq \mathbf{X}$ is Frontier-Reducible (FR) in $\mathcal{G}$ if $\exists X \in \mathbf{S}$ s.t. $\mathbf{S} \setminus \{X\} \in Fr_{\mathcal{G}}(X, Y)$.

In particular, if $\mathbf{S}$ is FR by $X \in \mathbf{S}$, $\nu(\mathbf{S}) = \nu(\mathbf{S} \setminus \{X\})$.

**Theorem D.9.** *Consider any FR $S \subseteq X$, and let us define $Z := \{X \in S \mid S \setminus \{X\} \in Fr_{\mathcal{G}}(X,Y)\} = \{X \in S \mid S_{>_{\mathcal{G}} X} \in Fr_{\mathcal{G}}(X,Y)\}$. Then $\nu(S) = \nu(S \setminus Z)$ and $S \setminus Z$ is not FR.*

*Proof.* Consider $Z = \{X_{i_1}, \ldots, X_{i_n}\}$ in the order $<_{\mathcal{G}}$. Note that $\forall j \in [n]$, $S \setminus \{X_{i_1}, \ldots, X_{i_j}\} = S_{>_{\mathcal{G}} X_{i_j}}$, which is a frontier between $X_{i_j}$ and $Y$ by construction.

Let us prove that $\forall j \in [n]$, $\nu(S) = \nu(S_{>_{\mathcal{G}} X_{i_j}})$ by induction. For $j = 1$, $\nu(S) = \nu(S \setminus \{X_{i_1}\})$ since $S_{>_{\mathcal{G}} X_{i_1}} \in Fr_{\mathcal{G}}(X_{i_1}, Y)$. For an arbitrary $j$, and assuming it true for $j - 1$, $\nu(S) = \nu(S_{>_{\mathcal{G}} X_{i_{j-1}}}) = \nu(S_{>_{\mathcal{G}} X_{i_j}})$, since $S_{>_{\mathcal{G}} X_{i_{j-1}}} \setminus \{X_{i_j}\} = S_{>_{\mathcal{G}} X_{i_j}} \in Fr_{\mathcal{G}}(X_{i_j}, Y)$. Therefore, for $j = n$, $\nu(S) = \nu(S \setminus \{X_{i_1}, \ldots, X_{i_n}\}) = \nu(S \setminus Z)$.

Additionally, $S \setminus Z$ is not FR since, if $\exists X \in S \setminus Z$ s.t. $S \setminus Z \setminus \{X\} \in Fr_{\mathcal{G}}(X,Y)$, then $S \setminus \{X\} \supseteq S \setminus Z \setminus \{X\}$ is also a frontier between $X$ and $Y$, which implies that $X \in Z$. $\square$

Finally, for a clearer characterization of the irreducible set, consider the following proposition.

**Proposition D.10.** *Given a FR $S \subseteq X$ and its corresponding irreducible $S' := S \setminus Z$, with $Z := \{X \in S \mid S_{>_{\mathcal{G}} X} \in Fr_{\mathcal{G}}(X,Y)\}$, then $S' = S \cap An_{\mathcal{G}_{\overline{S}}}(Y)$.*

*Proof.* We will show that $S \setminus Z = S \cap An_{\mathcal{G}_{\overline{S}}}(Y)$, or equivalently, that $\forall X \in S$, $S_{>_{\mathcal{G}} X} \notin Fr_{\mathcal{G}}(X,Y)$ iff $X \in An_{\mathcal{G}_{\overline{S}}}(Y)$.

Consider $X \in S$. If $S_{>_{\mathcal{G}} X} \notin Fr_{\mathcal{G}}(X,Y)$, then $S \setminus \{X\} \notin Fr_{\mathcal{G}}(X,Y)$ by Corollary D.7, so there is a directed path from $X$ to $Y$ not blocked by $S \setminus \{X\}$. Consequently, $X$ is an ancestor of $Y$ in the graph where we remove any incoming edges to $S$; in other words, $X \in An_{\mathcal{G}_{\overline{S}}}(Y)$. Conversely, if $X \in An_{\mathcal{G}_{\overline{S}}}(Y)$, there is a directed path from $X$ to $Y$ not blocked by $S \setminus \{X\}$, therefore $S \setminus \{X\} \notin Fr_{\mathcal{G}}(X,Y)$ and $S_{>_{\mathcal{G}} X} \notin Fr_{\mathcal{G}}(X,Y)$, again by Corollary D.7. $\square$

### D.2.2 ALGORITHM SOUNDNESS

Theorem D.9 identifies which elements can be removed from the computation of $\nu(S)$ for any set $S$. As a result, if we compute and cache $\nu(S \setminus Z)$, any other set with the same Frontier-Irreducible set can skip the $\nu$ computation and return the cached value instead. Additionally, we do not need to test identifiability for FR sets, only for the corresponding Frontier-Irreducible sets. We now need to define an efficient method to compute $S \setminus Z$, Algorithm 2, which consists of two procedures; let us first demonstrate the soundness of the Frontier-Reducibility Algorithm (FRA).

Given $S = (X_{i_1}, \ldots, X_{i_n})$ in $<_{\mathcal{G}}$ order, at step $k = n..1$, $X := X_{i_k}$ and $P := \{X_{i_n}, \ldots, X_{i_{k+1}}\} = S_{>_{\mathcal{G}} X_{i_k}}$. At this stage, we can check if $P \in Fr_{\mathcal{G}}(X,Y)$, or equivalently, if $P \cap De_{\mathcal{G}}(X) \in Fr_{\mathcal{G}}(X,Y)$, in which case we will include it in $Z$. At the end of the process, $Z = \{X \in S \mid S_{>_{\mathcal{G}} X} \in Fr_{\mathcal{G}}(X,Y)\}$, which, by Theorem D.9, means that $\nu(S) = \nu(S \setminus Z)$, and $S \setminus Z$ is not FR.

We include some optimizations to this algorithm. Firstly, we precompute `ParentsY`, `Descendants` and `Children` (the latter one for the `IS-FRONTIER` procedure) so that we do not need to traverse the graph every time they are needed. Secondly, we employ a cache for `Frontier`, which is populated as FRA processes more sets $S$. On the other hand, when checking if $P \in Fr_{\mathcal{G}}(X,Y)$, we check instead for $P' := P \cap De_{\mathcal{G}}(X)$, which is equivalent; this is so that we can better employ the `Frontier` cache, collapsing different $P \cup \{X\}$ sets into the same evaluation, with the added benefit that a lower number of parents when testing if a path is blocked by $P'$ will be faster. As a result, for any given set $S \subseteq X$, Algorithm 2 requires $|S|$ iterations (one per element of $S$), some of them skipped because $X \in Pa_{\mathcal{G}}(Y)$, some already cached in `Frontier`. Finally, this cache can be reused between explanations of do-SHAP for the same graph; only the $\nu$ cache must be reset every time. This speeds up further explanations with virtually zero cost from the FRA.

The next step is how to determine if a set $S \subseteq X \setminus \{X\}$ is a frontier between $X$ and $Y$. Naively, we could check if all directed paths between $X$ and $Y$ are blocked by (intersect with) $S$; we could precompute all paths and store them for faster access, but the number of paths grows exponentially (in the worst case scenario, i.e., a complete graph, there are $2^K - 1$ directed paths), which would in turn require an exponential number of iterations per frontier check. Instead, we devise a more efficient method, described in the `IS-FRONTIER` procedure in Algorithm 2.

**Algorithm 2** Frontier-Reducibility Algorithm: set version.

---

**Require:** ParentsY $:= Pa_{\mathcal{G}}(Y)$.
**Require:** $\forall X \in \mathbf{X}$, Descendants$[X] := De_{\mathcal{G}}(X)$.
**Require:** $\forall X \in \mathbf{X}$, Children$[X] := Ch_{\mathcal{G}}(X)$.
**Require:** Frontier, a map: tuple[int] $\rightarrow$ bool.

1: **procedure** IS-FRONTIER($\mathbf{S}, X, Y$, Children)
2:     $\mathbf{C} \leftarrow \{X\}$                                        ▷ Current nodes
3:     **while** $\mathbf{C} \neq \varnothing$ and $Y \notin \mathbf{C}$ **do**
4:         $\mathbf{S} \leftarrow \mathbf{S} \cup \mathbf{C}$
5:         $\mathbf{C} \leftarrow \bigcup_{C \in \mathbf{C}}$ Children$[C] \setminus \mathbf{S}$
6:     **end while**
7:     **return** $C = \varnothing$
8: **end procedure**
9: **procedure** FRA($\mathbf{S}$, ParentsY, Descendants, Children, Frontier)
10:     SORT($\mathbf{S}, <_{\mathcal{G}}$)                         ▷ Sort to move in descending order
11:     $\mathbf{P} \leftarrow \varnothing$
12:     $\mathbf{Z} \leftarrow \varnothing$
13:     $k \leftarrow |\mathbf{S}|$
14:     **while** $k > 0$ **do**
15:         $X \leftarrow \mathbf{S}[k]$                        ▷ $X = X_{i_k}$
16:         **if** $X \notin Pa_{\mathcal{G}}(Y)$ **then**
17:             $\mathbf{P'} \leftarrow \mathbf{P} \cap$ Descendants$[X]$     ▷ $\mathbf{P'} = \mathbf{S}_{>_{\mathcal{G}} X} \cap De_{\mathcal{G}}(X)$
18:             $\mathbf{T} \leftarrow \mathbf{P'} \cup \{X\}$
19:             **if** $\mathbf{T} \notin$ Frontier **then**
20:                 $v \leftarrow$ IS-FRONTIER($\mathbf{P'}, X, Y$, Children)
21:                 Frontier$[\mathbf{T}] \leftarrow v$
22:             **else**
23:                 $v \leftarrow$ Frontier$[\mathbf{T}]$
24:             **end if**
25:             **if** v **then**
26:                 $\mathbf{Z} \leftarrow \mathbf{Z} \cup \{X\}$
27:             **end if**
28:         **end if**
29:         $\mathbf{P} \leftarrow \mathbf{P} \cup \{X\}$
30:         $k \leftarrow k - 1$
31:     **end while**
32:     **return** $\mathbf{S} \setminus \mathbf{Z}$
33: **end procedure**

---

We now prove the validity of this procedure. Let us define $\mathbf{C}_0 := \{X\}$. $\mathbf{C}_0$ will never be empty nor contain $Y$, so we always enter the loop. At step $k = 1..n$, $\mathbf{S}_k := \bigcup_{k' < k} \mathbf{C}_{k'} \cup \mathbf{S}$ and $\mathbf{C}_k := \bigcup_{C \in \mathbf{C}_{k-1}} \texttt{Children}[C] \setminus \mathbf{S}_k$. All directed paths from $X$ to $Y$ are sequences of parent-child pairs, just as any node $C \in \mathbf{C}_k$ is a child of a certain node $C' \in C_{k-1}$. Additionally, since every node in $\mathbf{X}$ is an ancestor of $Y$, by exploring these parent-child sequences we will necessarily result in a directed path from $X$ to $Y$. Therefore, every directed path is covered by a sequence of nodes $C_k \in \mathbf{C}_k$ unless they are discarded by $\mathbf{S}_k$, in which case either $\mathbf{S}$ blocked the node in the path, or it was a node already visited before which would continue with a subpath $C \to \cdots \to Y$ that is currently being explored or has already been discarded.

Note that since $\mathbf{S}_k$ removes any already-visited nodes from $\mathbf{C}_k$, and we always move one level deeper in the graph, $\mathbf{C}_k$'s nodes are all necessarily at depth $k$ from $X$. Given that the graph $\mathcal{G}$ is finite and acyclic, $\mathbf{C}$ will eventually be empty or contain $Y$ (since it is the last node in any path), which guarantees that the loop ends. Let $\mathbf{C}_n$ denote the last step. Note that if $\mathbf{C}_n \neq \varnothing$, then $Y \in \mathbf{C}_n$, which means that there was a sequence of nodes, each a child of the previous one, that were never filtered by $\mathbf{S}$; in other words, there exists a directed path from $X$ to $Y$ that is not blocked by $\mathbf{S}$, therefore $\mathbf{S} \notin Fr_{\mathcal{G}}(X, Y)$, and the procedure returns false. On the other hand, if $\mathbf{C}_n = \varnothing$, then every sequence of nodes (every path) was eventually blocked by $\mathbf{S}$; therefore, $\mathbf{S} \in Fr_{\mathcal{G}}(X, Y)$, and the procedure returns true.

In terms of execution time, since every step results in nodes one depth level deeper, the number of iterations of this procedure cannot be higher than the maximum depth of the graph, which, in the worst case scenario (e.g., a chain graph) is $K$, making it much more efficient than the naive strategy. We incorporate this procedure as part of the larger FRA algorithm.

### D.2.3 INTEGER FORMULATION

We can further optimize this algorithm by transforming set-operations into integer and binary operations, resulting in the algorithm presented in the main paper, repeated in Algorithm 3 for the reader's convenience. Let us demonstrate that both algorithms are equivalent. Given $\mathbf{X} = (V_0, \ldots, V_{K-1})$, $K := |\mathbf{X}|$, there is a bijection $\phi : \mathbb{P}(\mathbf{X}) \to \{0, \ldots, 2^K - 1\}$ such that $\phi(\mathbf{S}) = \sum_{V_k \in \mathbf{S}} 2^k$. Note that $\phi(\mathbf{S})$ is a $K$-length binary array with 1s in each position $k$ (starting from the end) such that $V_k \in \mathbf{S}$. Consequently, let us define, for any $s \in \{1, \cdots, 2^K - 1\}$, $\psi(s) := \lfloor \log_2 s \rfloor$; then $\psi(s) = \max \{k \mid V_k \in \phi^{-1}(s)\}$; if we subtract $2^{\psi(s)}$ from $s$, we can apply $\psi$ again to retrieve the second-largest element, and so on until $s = 0$ ($\mathbf{S} = \varnothing$). The sequence of elements $\psi(s)$ returns the original set $\mathbf{S} = \phi^{-1}(s)$.

Thanks to this bijection, we can perform all our operations directly on integers with arithmetic and binary operations, which are less expensive, timing- and memory-wise, than with operations over sequences of integers. Note that, $\forall \mathbf{S}, \mathbf{S'} \subseteq \mathbf{X}$:

1. $\phi(\mathbf{S} \cap \mathbf{S'}) = \phi(\mathbf{S}) \ \& \ \phi(\mathbf{S'})$, with $\&$ the bitwise AND operator.
2. $\phi(\mathbf{S} \cup \mathbf{S'}) = \phi(\mathbf{S}) \mid \phi(\mathbf{S'})$, with $\mid$ the bitwise OR operator.
3. $\phi(\mathbf{S} \setminus \mathbf{S'}) = \phi(\mathbf{S}) \ \& \ \neg \phi(\mathbf{S'})$, with $\neg$ the bitwise NOT operator.
4. $\mathbf{S} \cap \mathbf{S'} = \varnothing \Rightarrow \phi(\mathbf{S} \cup \mathbf{S'}) = \phi(\mathbf{S}) + \phi(\mathbf{S'})$.
5. $\mathbf{S} \supseteq \mathbf{S'} \Rightarrow \phi(\mathbf{S} \setminus \mathbf{S'}) = \phi(\mathbf{S}) - \phi(\mathbf{S'})$.

Let us compare between Algorithms 2 and 3. Firstly, we precompute and $\phi$-encode `ParentsY`, `Descendants` and `Children`, since they will be used repeatedly throughout the algorithm. Note that we do not need to sort $\mathbf{S}$ beforehand, instead passing it in its $s := \phi(\mathbf{S})$ representation [7]. We can obtain the elements $x$ in descending order, already encoded, by computing $x := 2^{\lfloor \log_2 s \rfloor}$ and subtracting it from $s$. We can check if an encoded $x$ is a parent of $Y$ with the AND operator. We can restrict $\mathbf{P}$, encoded by an integer $p$, to $\mathbf{P'} := \mathbf{P} \cap De_{\mathcal{G}}(X)$, encoded by an integer $p'$, by using the AND operator on the precomputed encoded set `ParentsY` $:= \phi(Pa_{\mathcal{G}}(Y))$. We can identify the cache-key $\mathbf{P'} \cup \{X\}$ by its code $p' + x$. In terms of the Is-FRONTIER procedure, we iterate over the elements in $\mathbf{C}_k$ by employing the same strategy as before.

---

[7]It is more efficient to pass $s$ directly to the FRA procedure, since we can pre-encode all indices $\{0, \ldots, K-1\}$ before generating permutations of them; then, we just need to pass the sum of the chosen coalition.

---

**Algorithm 3** Frontier-Reducibility Algorithm (FRA)

---

**Require:** $\texttt{ParentsY} := \sum_{V_k \in Pa_{\mathcal{G}}(Y)} 2^k$.

**Require:** $\forall k = 0..K-1, \texttt{Descendants}[2^k] := \sum_{V_{k'} \in De_{\mathcal{G}}(V_k)} 2^{k'}$.

**Require:** $\forall k = 0..K-1, \texttt{Children}[2^k] := \sum_{V_{k'} \in Ch_{\mathcal{G}}(V_k)} 2^{k'}$.

**Require:** $\texttt{Frontier}$, a map int $\to$ bool.

1: **procedure** IS-FRONTIER($s, x, y, \texttt{Children}$)
2:      $c \leftarrow x$               $\triangleright$ Current nodes
3:      **while** $c \neq 0$ and $y$ & $c = 0$ **do**     $\triangleright$ While $\mathbf{C} \neq \varnothing$ and $Y \notin \mathbf{C}$
4:          $s \leftarrow s \mid c$        $\triangleright$ Update visited nodes with the new nodes
5:          $c' \leftarrow c$
6:          **while** $c' > 0$ **do**        $\triangleright$ Iterate over the elements in $\mathbf{C}$
7:             $x \leftarrow 2^{\lfloor \log_2 c' \rfloor}$
8:             $c' \leftarrow c' - x$
9:             $c \leftarrow c \mid \texttt{Children}[x]$
10:         **end while**
11:         $c \leftarrow c$ & $\neg s$        $\triangleright$ Remove any previously visited nodes
12:      **end while**
13:      **return** c $= 0$
14: **end procedure**
15: **procedure** FRA($s, \texttt{ParentsY}, \texttt{Descendants}, \texttt{Children}, \texttt{Frontier}$)
16:      $p \leftarrow 0; z \leftarrow 0;$          $\triangleright$ Initialize $\mathbf{P}, \mathbf{Z}$ (encoded)
17:      **while** $s > 0$ **do**
18:          $x \leftarrow 2^{\lfloor \log_2 s \rfloor}$        $\triangleright$ Get the last element (encoded)
19:          $s \leftarrow s - x$
20:          **if** $x$ & $\texttt{ParentsY} = 0$ **then**     $\triangleright$ Only if not a parent of $Y$
21:             $p' \leftarrow p$ & $\texttt{Descendants}[x]$    $\triangleright$ Only check descendants
22:             $t \leftarrow p' + x$      $\triangleright$ $p' + x$ uniquely defines $(p', x)$
23:             **if** $t \notin \texttt{Frontier}$ **then**
24:                 $v \leftarrow$ IS-FRONTIER($p', x, y, \texttt{Children}$)
25:                 $\texttt{Frontier}[t] \leftarrow v$
26:             **else**
27:                 $v \leftarrow \texttt{Frontier}[t]$
28:             **end if**
29:             **if** $v$ **then**
30:                 $z \leftarrow z + x$
31:             **end if**
32:          **end if**
33:          $p \leftarrow p + x$
34:      **end while**
35:      **return** $p - z$         $\triangleright$ Return the encoded set $\mathbf{S} \setminus \mathbf{Z}$
36: **end procedure**

---

All of these changes result in an equivalent algorithm, more time-efficient (as demonstrated in Appendix E.2) and memory-efficient (since we operate and cache integers rather than sets of integers).

### D.3 do-Shapley value for the noise

*Remark* D.11. **No Bow Patterns**.
Given a target variable $Y$, if we consider the SCM projected to the ancestors of $Y$, with $\mathbf{X}$ denoting the set of features in this graph, and assume that do-SHAP is identifiable in this graph, then, $\forall X \in \mathbf{X}$, there are no *bow patterns* from $X$ to $Y$ (a back-door path $X \leftarrow \cdots \to \cdots \overset{?}{\text{---}} Y$ consisting only of latent nodes between $X$ and $Y$). If there were, the query $\nu_{\mathbf{x}}(\{X\}) = \mathbb{E}[Y \mid do(X = x)]$ would not be identifiable (Pearl, 2009) (section 3.5.2).

**Theorem D.12.** *do-Shapley Value for the Noise*.
*Given a target r.v. $Y \in \mathcal{V}$, consider the projected SCM $\mathcal{M}[An(Y)]$, with $\mathbf{X} := An(Y) \setminus \{Y\}$ and*

*realizations $(\boldsymbol{x}, y) \sim P(\boldsymbol{X}, Y)$. Let $(\phi_X := \phi_{\nu_x}(X))_{X \in \boldsymbol{X}}$ be the (identifiable) do-Shapley values associated with $K$ players $\boldsymbol{X}$.*

*Let us assume that $f_Y \in \mathcal{F}$ follows an additive noise model, i.e., $Y = f_Y(Pa_Y, E_Y) = f(Pa_Y) + E_Y$ for an unknown function $f$. Let $\phi'$ be the do-Shapley values w.r.t. players $\boldsymbol{X}' := \boldsymbol{X} \cup \{E_Y\}$; then, for any $X \in \boldsymbol{X}$, $\phi'_X = \phi_X$ and $\phi'_{E_Y} = y - \mathbb{E}[Y \mid pa_Y]$. Furthermore, $\sum_{X \in \boldsymbol{X}} \phi'_X + \phi'_{E_Y} = y - \mathbb{E}[Y]$.*

*Proof.* Let us define some notation for convenience:

- $\forall \mathbf{S} \subseteq \mathbf{X}$, let $\mathbf{S}^c := \mathbf{X} \setminus \mathbf{S}$.

- Note that $Pa_Y \subseteq \mathbf{S} \cup \mathbf{S}^c$; let us denote the selected values $pa_Y$ as the output of a function $Pa_Y(\mathbf{s}, \mathbf{s}^c)$ for ease of exposition.

- Let $\nu'(\mathbf{S}) := \mathbb{E}_{\mathbf{S}^c | \widehat{\mathbf{s}}}[f(Pa_Y(\mathbf{s}, \mathbf{S}^c))]$ for convenience of notation.

We want to compute do-Shapley values $\phi'$ for the $(K+1)$-game (including $E_Y$) with realizations $(\varepsilon_Y, \mathbf{x}, y) \sim \mathcal{P}(E_Y, \mathbf{X}, Y)$ (with $\varepsilon_Y$ latent, unknown) based on the values $\phi$ for the $K$-game (only including $\mathbf{X}$) with the same realizations $(\mathbf{x}, y) \sim \mathcal{P}(\mathbf{X}, Y)$. Let us first determine the value of the following two quantities for any $\mathbf{S} \subseteq \mathbf{X}$ ($E_Y \notin \mathbf{S}$):

$$\begin{aligned}
\nu(\mathbf{S} \cup \{E_Y\}) &= \mathbb{E}[Y \mid \widehat{\mathbf{s}}, \widehat{\varepsilon_Y}] \\
&= \mathbb{E}_{\mathbf{S}^c | \widehat{\mathbf{s}}, \widehat{\varepsilon_Y}}[Y \mid \widehat{\mathbf{s}}, \mathbf{S}^c, \widehat{\varepsilon_Y}] \\
&= \mathbb{E}_{\mathbf{S}^c | \widehat{\mathbf{s}}}[f(Pa_Y(\mathbf{s}, \mathbf{S}^c))] + \varepsilon_Y \\
&= \nu'(\mathbf{S}) + \varepsilon_Y.
\end{aligned} \tag{9}$$

We can perform the first step by marginalizing over $\mathbf{S}^c$ in $\mathcal{P}_{\mathbf{s}, \varepsilon_Y}$. Then, $(\mathbf{S}^c \perp\!\!\!\perp E_Y \mid \mathbf{S})_{\mathcal{G}_{\overline{\mathbf{s}, E_Y}}}$ because any path $p$ connecting $E_Y$ must necessarily have a collider in $Y$, since $De(Y) = Y$, therefore blocking the path. By R3, $\mathcal{P}_{\widehat{\mathbf{s}}, \widehat{\varepsilon_Y}}(\mathbf{S}^c) = \mathcal{P}_{\widehat{\mathbf{s}}}(\mathbf{S}^c)$ so we can remove it from the expectation over $\mathbf{S}^c$. On the other hand, we know that $Y = f(Pa_Y) + E_Y$ and by the linearity of expectations, we can remove $\varepsilon_Y$ from the expectation. Finally, for later clarity, we can denote the first term by $\nu'(\mathbf{S})$.

Next, let us solve the analogous term for $\mathbf{S}$:

$$\begin{aligned}
\nu(\mathbf{S}) &= \mathbb{E}[Y \mid \widehat{\mathbf{s}}] \\
&= \mathbb{E}_{\mathbf{S}^c, E_Y | \widehat{\mathbf{s}}}[Y \mid \widehat{\mathbf{s}}, \mathbf{S}^c, E_Y] \\
&= \mathbb{E}_{\mathbf{S}^c | \widehat{\mathbf{s}}}[f(Pa_Y(\mathbf{s}, \mathbf{S}^c))] + \mathbb{E}[E_Y] \\
&= \nu'(\mathbf{S}) + \mathbb{E}[E_Y].
\end{aligned} \tag{10}$$

We proceed similarly, beginning with a marginalization over $\mathbf{S}^c$ and $E_Y$ this time. In order to separate $E_Y$ from the first term, note that $(\mathbf{S}^c \perp\!\!\!\perp E_Y)_{\mathcal{G}_{\overline{\mathbf{s}}}}$ and $(\mathbf{S} \perp\!\!\!\perp E_Y)_{\mathcal{G}_{\overline{\mathbf{s}}}}$ for the same reason as before: any path connecting $E_Y$ must necessarily pass through $Y$, which acts as an unconditioned collider, thereby blocking it. Thanks to these two properties, $P(\mathbf{S}^c, E_Y \mid \widehat{\mathbf{s}}) = P(\mathbf{S}^c \mid \widehat{\mathbf{s}}) P(E_Y \mid \widehat{\mathbf{s}}) = P(\mathbf{S}^c \mid \widehat{\mathbf{s}}) P(E_Y)$, with the former step by the rules of independence and the latter by R3. We can then split the expectation, this time with $\mathbb{E}[E_Y]$ as the second term and using the same notation $\nu'(\mathbf{S})$ again.

With these two computations, we can see that:

$$\nu(\mathbf{S} \cup \{E_Y\}) - \nu(\mathbf{S}) = \varepsilon_Y - \mathbb{E}[E_Y], \tag{11}$$

and substituting it into the SHAP formula (with $K+1$ players):

$$\phi'_{E_Y} = \sum_{\mathbf{S} \subseteq \mathbf{X}} \frac{1}{K+1} \binom{K}{|S|}^{-1} (\varepsilon_Y - \mathbb{E}\left[E_Y\right])$$

$$= \sum_{s=0}^{K} \binom{K}{s} \frac{1}{K+1} \binom{K}{s}^{-1} (\varepsilon_Y - \mathbb{E}\left[E_Y\right])$$

$$= \varepsilon_Y - \mathbb{E}\left[E_Y\right]. \tag{12}$$

We transform the first to second step by realizing that we do not need to know what the coalitions $\mathbf{S}$ are, only their cardinality, so we can transform $\sum_{\mathbf{S} \subseteq \mathbf{X}}$ into $\sum_{s=0}^{K}$ by multiplying by $\binom{K}{s}$, the number of combinations of $s$ elements. This term and its inverse cancel out, and $K+1$ constant terms summed together cancels with $\frac{1}{K+1}$, resulting in $\varepsilon_Y - \mathbb{E}\left[E_Y\right]$.

Now, note that $\varepsilon_Y = y - f(pa_Y)$ and:

$$\mathbb{E}\left[Y \mid pa_Y\right] = \mathbb{E}\left[f(pa_Y) + E_Y\right] = f(pa_Y) + \mathbb{E}\left[E_Y\right], \tag{13}$$

so $f(pa_Y) = \mathbb{E}\left[Y \mid pa_Y\right] - \mathbb{E}\left[E_Y\right]$. Then,

$$\phi'_{E_Y} = \varepsilon_Y - \mathbb{E}\left[E_Y\right] = y - f(pa_Y) - \mathbb{E}\left[E_Y\right] = y - \mathbb{E}\left[Y \mid pa_Y\right]. \tag{14}$$

This proves the value for $\phi'_{E_Y}$. Let us now compute $\phi'_X$ for any $X \in \mathbf{X}$. Let $\mathbf{S} \subseteq (\mathbf{X} \cup \{E_Y\}) \setminus \{X\}$. If $E_Y \in \mathbf{S}$, we can apply Equation (9) and $\nu(\mathbf{S}) = \nu'(\mathbf{S} \setminus \{E_Y\}) + \varepsilon_Y$. Otherwise, Equation (10) gives us $\nu(\mathbf{S}) = \nu'(\mathbf{S}) + \mathbb{E}\left[E_Y\right]$. Now, $\phi'_X$'s computation can use both results:

$$\phi'_X = \sum_{\mathbf{S} \subseteq \mathbf{X} \setminus \{X\}} \frac{1}{K+1} \binom{K}{|\mathbf{S}|+1}^{-1} (\nu(\mathbf{S} \cup \{E_Y, X\}) - \nu(\mathbf{S} \cup \{E_Y\}))$$

$$+ \sum_{\mathbf{S} \subseteq \mathbf{X} \setminus \{X\}} \frac{1}{K+1} \binom{K}{|\mathbf{S}|}^{-1} (\nu(\mathbf{S} \cup \{X\}) - \nu(\mathbf{S}))$$

$$= \sum_{\mathbf{S} \subseteq \mathbf{X} \setminus \{X\}} \frac{1}{K+1} \left( \binom{K}{|\mathbf{S}|+1}^{-1} (\nu(\mathbf{S} \cup \{X\}) - \nu(\mathbf{S})) + \binom{K}{|\mathbf{S}|}^{-1} (\nu(\mathbf{S} \cup \{X\}) - \nu(\mathbf{S})) \right)$$

$$= \sum_{\mathbf{S} \subseteq \mathbf{X} \setminus \{X\}} \frac{1}{K+1} \left( \binom{K}{|\mathbf{S}|+1}^{-1} + \binom{K}{|\mathbf{S}|}^{-1} \right) (\nu(\mathbf{S} \cup \{X\}) - \nu(\mathbf{S}))$$

$$= \sum_{\mathbf{S} \subseteq \mathbf{X} \setminus \{X\}} \frac{1}{K} \binom{K-1}{|\mathbf{S}|}^{-1} (\nu(\mathbf{S} \cup \{X\}) - \nu(\mathbf{S})) = \phi_X \tag{15}$$

We first split the SHAP formula in two: those sets that include $E_Y$ and those that do not; note that the combination terms are altered to reflect the size of the base set applied to $\nu$ ($|\mathbf{S}|+1$ in the first case since the base set is $\mathbf{S} \cup \{E_Y\}$). For the first to second step, we can bring together the two sums, and transform the first difference,

$$\nu(\mathbf{S} \cup \{E_Y, X\}) - \nu(\mathbf{S} \cup \{E_Y\}) = \nu'(\mathbf{S} \cup \{X\}) - \nu'(\mathbf{S}) = \nu(\mathbf{S} \cup \{X\}) - \nu(\mathbf{S}), \tag{16}$$

by applying Equation (9) and Equation (10) and cancelling the $\varepsilon_Y$ and $\mathbb{E}\left[E_Y\right]$ terms, respectively. We now sum the two inverse combination terms; let $s := |\mathbf{S}|$, with $s \leq K - 1$ since $\mathbf{S} \subseteq \mathbf{X} \setminus \{X\}$:

$$\binom{K}{s+1}^{-1} + \binom{K}{s}^{-1} = \frac{(s+1)!(K-s-1)! + s!(K-s)!}{K!}$$

$$= \frac{(s+1) \cdot s!(K-s-1)! + (K-s) \cdot s!(K-s-1)!}{K \cdot (K-1)!}$$

$$= \frac{(s+1+K-s)s!(K-s-1)!}{K \cdot (K-1)!}$$

$$= \frac{K+1}{K} \cdot \frac{s!(K-s-1)!}{(K-1)!} = \frac{K+1}{K}\binom{K-1}{s}^{-1} \tag{17}$$

Substituting this result back into Equation (15), we can cancel out $K + 1$. We arrive at the last formula, which is exactly the value for do-SHAP in the $K$-player game without $E_Y$.

Finally, to prove the last statement, $\sum_{X \in \mathbf{X}} \phi'_X + \phi'_{E_Y} = y - \mathbb{E}[Y]$, let us discuss two facts. Firstly, as stated in Remark D.11, there are no bow patterns between any $X \in \mathbf{X}$ and $Y$, therefore, no latent confounders connected to $Y$. Secondly, let us prove that $\mathbb{E}[Y \mid \widehat{\mathbf{x}}] = \mathbb{E}[Y \mid pa_Y]$:

$$\mathbb{E}[Y \mid \widehat{\mathbf{x}}] = \mathbb{E}\left[Y \mid \widehat{pa_Y}, \widehat{pa_Y^c}\right] = \mathbb{E}[Y \mid \widehat{pa_Y}] = \mathbb{E}[Y \mid pa_Y]. \tag{18}$$

We first split $\mathbf{X} = Pa_Y \cup Pa_Y^c$. Next, we apply R3 with $(Y \perp\!\!\!\perp Pa_Y^c \mid Pa_Y)_{\mathcal{G}_{\overline{\mathbf{X}}}}$ since only the edges ending in $Y$ remain in $\mathcal{G}_{\overline{\mathbf{X}}}$, which are the ones starting on $Pa_Y$ and the ones connected to confounders $\mathcal{U}_{\{Y,\cdot\}}$, but there are none. No matter the case, $Pa_Y^c$ is not connected through any of these paths, so it is independent of $Y$. Then, we apply R2 with $(Y \perp\!\!\!\perp Pa_Y)_{\mathcal{G}_{\underline{Pa_Y}}}$: since $Y$ has no descendants and we remove any incoming edges to $Y$ in $\mathcal{G}_{\underline{Pa_Y}}$ except for the ones starting from confounders in $\mathcal{U}_{\{Y,\cdot\}}$, of which there are none, $Y$ must be independent of $Pa_Y$, proving the expression.

Now we can prove the remaining fact:

$$\sum_{X \in \mathbf{X}} \phi'_X + \phi'_{E_Y} = (\mathbb{E}[Y \mid \widehat{\mathbf{x}}] - \mathbb{E}[Y]) + (y - \mathbb{E}[Y \mid pa_Y]) = y - \mathbb{E}[Y] \tag{19}$$

$\square$

# E EXPERIMENTS

These experiments are executed on personal computers (particularly, a Macbook with an M3 Pro chip) and do not require an infrastructure of workers for their execution. No experiment lasted longer than 6h to execute, reason why we did not take measurements of their times.

## E.1 SYNTHETIC DATASET

We include here further details about the Synthetic experiment in Section 5.1, bigger figures for the Markovian case (for better visibility) and a discussion on the semi-Markovian case. Please refer to the supplementary code for the actual implementation of these experiments, available with the submission of the camera-ready version of this work.

### E.1.1 IMPLEMENTATION DETAILS

We chose several SCM architectures for the experiment; here we justify these choices. Regarding the classic approach of modeling an SCM with each of its functions $f_k \in \mathcal{F}$ separately, some of the approaches mentioned in Section 2 focus on how to model complex multivariate r.v.s (e.g., images), but the remaining univariate r.v.s are modelled by specifying which probability distribution family they belong to. Since our DGP consists exclusively of univariate continuous r.v.s, these proposals are equivalent. Instead, we will employ Deep Causal Graph (DCG) (Parafita & Vitrià, 2022), a general framework for all sorts of implementations of SCMs. In particular, with DCGs, we can train

three different kinds of SCM: 1) a **linear** SCM with Normal distributions for each variable, used as a baseline; 2) the Distributional Causal Node (Parafita & Vitrià, 2019) architecture, where every node is modeled after a probability distribution family with a feed-forward network for the computation of its parameters (**DCN**); and 3) **DCG**s powered with its most flexible implementation for continuous nodes, based on Normalizing Flows. Finally, in order to test the alternative approach of modeling SCMs not node-wise, but the graph as a whole, we could use Variational Causal Autoencoder (VACA) (Sánchez-Martín et al., 2022) or Causal Normalizing Flows (**CNF**) (Javaloy et al., 2024). However, as stated by the authors of CNF based on their experiments, "VACA shows poor performance, and is considerably slower due to the complexity of GNNs". For this reason, we opt for CNFs as a representative of this alternative approach for SCM modeling.

Regarding the definition of the synthetic DGP, we employ a set of non-linear functions along with exogenous samples from diverse continuous r.v.s for the generation of new samples. Let $\chi^2(k)$ be the Chi-squared distribution with $k$ degrees of freedom, $\mathcal{B}(\alpha, \beta)$ the Beta distribution, $\mathcal{N}(\mu, \sigma)$ the Normal distribution, and $\texttt{Exponential}(\lambda)$ the Exponential distribution. We sample from each latent variable first and then apply the functions in $\mathcal{F}$ in topological order:

$$
\begin{cases}
u \sim \chi^2(k = 10); \\
\varepsilon_Z \sim \mathcal{B}(\alpha = 2, \beta = 5); \\
\varepsilon_X \sim \mathcal{N}(\mu = 0, \sigma = 0.1); \\
\varepsilon_{A,1} \sim \texttt{Exponential}(\lambda = 1); \\
\varepsilon_{A,2} \sim \mathcal{N}(\mu = 0, \sigma = 0.1); \\
\varepsilon_B \sim \mathcal{N}(\mu = 0, \sigma = 1); \\
\varepsilon_C \sim \mathcal{N}(\mu = 0, \sigma = 0.5); \\
\varepsilon_Y \sim \mathcal{N}(\mu = 0, \sigma = 0.5);
\end{cases}
\qquad
\begin{cases}
z \leftarrow \varepsilon_Z; \\
x \leftarrow |z(u - 5) + \varepsilon_X|; \\
a \leftarrow |\sqrt{x} + \varepsilon_{A,1} + \varepsilon_{A,2}|; \\
b \leftarrow 5\sin(a) - \frac{u}{10} + \varepsilon_B; \\
c \leftarrow \log(1 + b^2) + \varepsilon_C; \\
y \leftarrow \log\frac{z}{1-z} + \left(\frac{x}{10}\right)^2 + c + \varepsilon_Y;
\end{cases}
\tag{20}
$$

Note that we have diverse continuous variables: some non-negative, some restricted to $(0, 1)$, some with unrestricted support. For the **DCN** implementation, where we need to assume the r.v.'s probability distribution family, we will employ Exponential, Beta and Normal distributions respectively. Each set of parameters $\Theta_X$ can be computed with a shared feed-forward network using a Graphical Conditioner (Parafita & Vitrià, 2022). For the **DCG** implementation with Normalizing Flows, we will use a Normal Distribution as its base noise ($E_X$) and the following flow structure (from $X$ to $E_X$):

- Affine layer, $f(x) = \sigma x + \mu$, with $\mu, \sigma$ learnable parameters.
- 3 blocks of:
  - Rational-Quadratic Spline Flow (Durkan et al., 2019), defined on the interval $[-5, 5]$, with $K = 8$ bins.
  - Affine layer.

Additionally, depending on the support of the r.v. to be modelled, we prepend another layer to transform the original domain into $\mathbb{R}$ before the application of the first Affine layer. In particular, for flows defined in $(0, 1)$, we use a Logit transformation $f(x) := \log\frac{x}{1-x}$, and for non-negative flows, an inverse Softplus layer $f(x) := \log(e^x - 1)$ (using the identity after a certain threshold for numerical stability). All parameters not set as hyperparameters are computed by an external trainable Conditioner that takes the node's parents as input and outputs their value.

Regarding the Conditioner network's architecture, it is a standard feed-forward network with ELU activations (Clevert et al., 2015), 2 hidden layers of dimension 32, and a standardizer layer at the beginning defined with the training dataset. This is used for the **DCN** and **DCG** SCMs. Regarding the **CNF** architecture, it uses a Softclip-constrained NSF-based architecture similar to ours, but with 4 stacked Spline layers (the diameter of the graph) and a MADE Conditioner to learn to model all variables at once. In this case, the conditioner uses 3 hidden layers with dimension 32 and ELU as its activation.

In terms of training, we use the AdamW optimizer (Loshchilov & Hutter, 2019) with Early Stopping (after 100 epochs with no improvement), using learning rate $10^{-3}$, weight decay $10^{-2}$ and batch

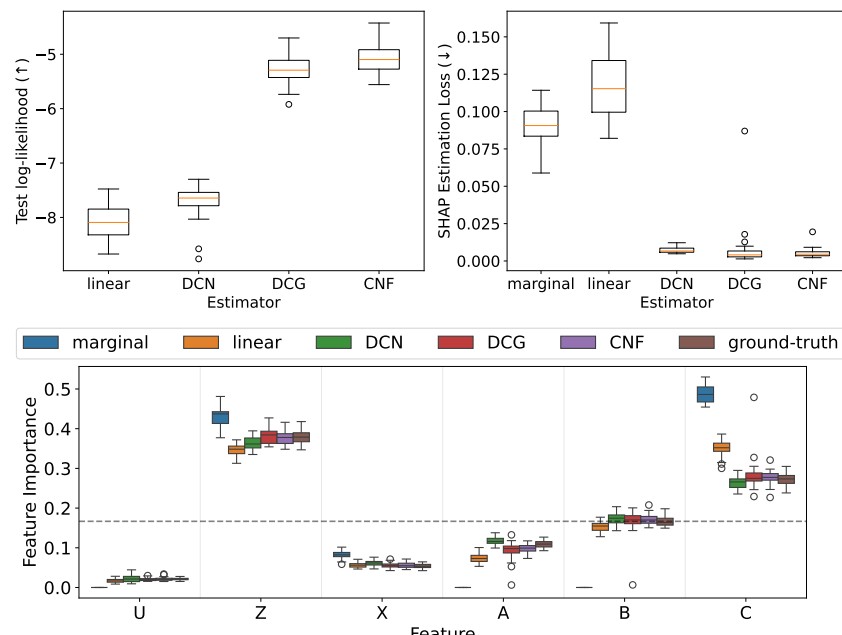

Figure 6: Markovian case. Box-plots computed over 30 realizations of the dataset. (a) Distribution adjustment score, log-likelihood (bigger is better). (b) SHAP estimation loss, $\mathcal{L}$ (lower is better). (c) Feature Importance (the closer to *ground-truth*, the better). Dashed horizontal line represents uniform importance ($\frac{1}{K}$).

size 100. Regarding SHAP estimation, since we only have 5 variables, we use the exact permutation method, taking 1,000 samples from each SCM for the Monte Carlo estimators of $\nu(\mathbf{S})$.

Finally, see Figure 6 for a bigger representation of the plots in the Markovian case.

### E.1.2 SEMI-MARKOVIAN CASE

We also test our approach on the semi-Markovian case, where the latent confounder $U_{\{X,B\}}$ is not observed. As stated in Section 5.1, CNF cannot be applied without the causal sufficiency assumption, so we will proceed with the remaining SCMs.

Figure 7 shows the same three plots as in the Markovian case, with similar results. The linear SCM cannot properly estimate $\mathcal{P}(\mathcal{V})$, which results in worse SHAP loss. DCNs achieve similar results to DCGs, but DCGs exhibit the best distribution adjustment and estimation performance. Finally, marginal SHAP results in different estimations than do-SHAP, something that is patently clear in the FI plot, where $Z$ and $C$'s contribution are overestimated while $A$ and $B$ are underestimated. Note that we obtain the same explanations as in the Markovian case even though we cannot measure the latent confounder nor know its distribution.

### E.2 FRA EXPERIMENTS

We will now describe additional tests for FRA.

Figure 8 shows comparisons between both versions of the FRA algorithm: FR1 (using sets, Algorithm 2) and FR2 (using integers, Algorithm 3). Note that FR2 is consistently and significantly faster that FR1, and less memory-intensive, since the `Frontier` cache needs only store numerical encodings of sets, instead of sets of integers. However, depending on graph topology, particularly if the number of edges is too high ($p = 0.9$) FR1 can be faster than FR2, as shown by Figure 8 (b), possibly because the procedure to iterate through $\mathbf{C}$ in the IS-FRONTIER procedure of FR2 is more expensive than in FR1. In any case, FRA is negligible time-wise *w.r.t.* the time needed for the estimation of $\nu(\mathbf{S})$, as we will see next.

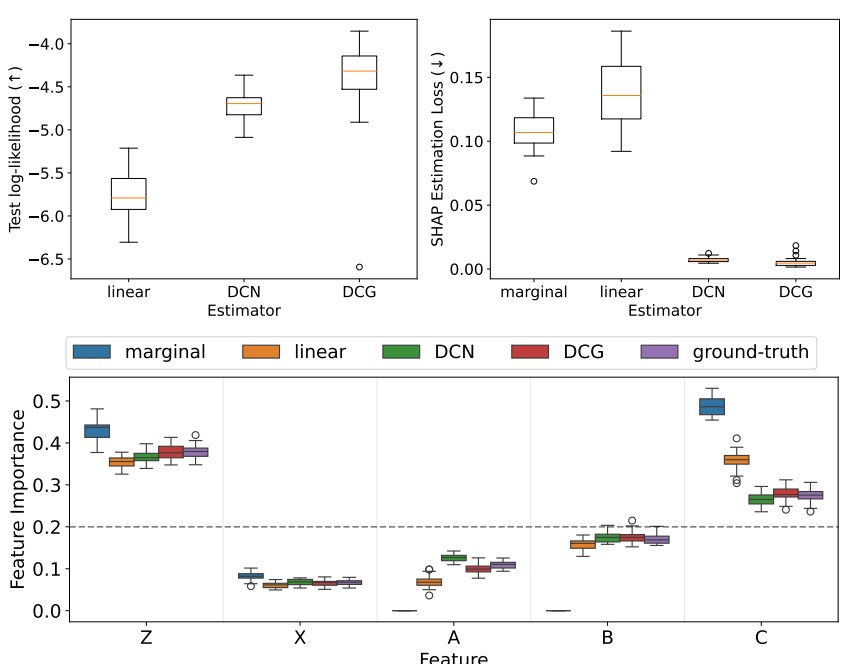

Figure 7: Semi-Markovian case. Box-plots computed over 30 realizations of the dataset. (a) Distribution adjustment score, log-likelihood (bigger is better). (b) SHAP estimation loss, $\mathcal{L}$ (lower is better). (c) Feature Importance (the closer to ground-truth, the better). Dashed horizontal line represents uniform importance $\left(\frac{1}{K}\right)$.

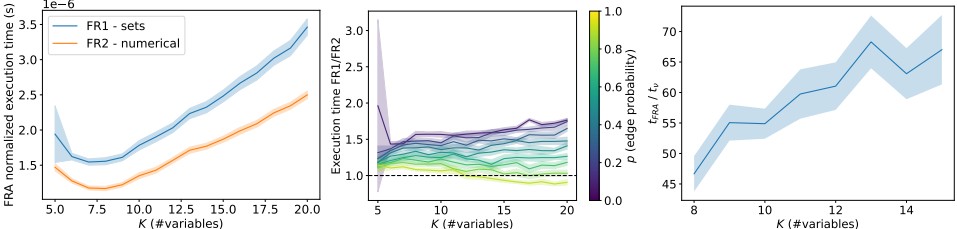

Figure 8: FRA experiments. (a) shows the errors bars (at 2-sigma over 270 replications) for the average normalized execution times of FR1 (sets) and FR2 (numerical) per number of nodes ($K$). (b) shows the error bars (at 2-sigma over 30 replications) of the ratio between FR1 time and FR2 time, split by edge probability ($p$), over the number of nodes $K$. If above 1, FR2 is faster. (c) shows the error bars (at 2-sigma over 30 replications) for the mean of quotients between the total time executing FRA (+permutations) and for estimating all sampled $\nu(\mathbf{S})$.

In the second experiment, as described in Section 5.2, we employed a synthetic DGP consisting of a linear function. We generate 1,000 i. i. d. samples from this DGP to create training, validation and test sets with ratios 8:1:1. We train a linear DCG with Normal DCN nodes, which is fitting for this dataset and deliberately lightweight, so as to show the improvements from the application of FRA even in the case where the model is particularly fast; FRA can only improve time-gains with more expressive and expensive models. We use a learning rate of $10^{-2}$, weight decay of $10^{-2}$, a batch size of 100, early stopping with patience of 10 epochs and AdamW (Loshchilov & Hutter, 2019) as the optimizer. For each $\nu$ estimation, we generate 100 samples from the SCM.

For this experiment, we kept track of do-SHAP execution time as well as the time for the computation of $\nu(\mathbf{S})$ specifically; this allows us, by subtraction, to compute the time needed for FRA and generating the permutations $\pi$ from where the sets $\mathbf{S}$ emerge. If we compare these two quantities, dividing the $\nu$ execution time by the FRA+permutation time, as shown in Figure 8 (c), we see that FRA is orders of magnitude faster than $\nu$; this means that, at a negligible cost, we can speed up do-SHAP by a significant ratio, specifically the number of FR coalitions.

# F APPLICATIONS

In this section, we showcase the application of do-SHAP explanations on two real-world datasets as illustrative examples of the explanatory power of our techniques.

## F.1 DIABETES DATASET

Here we discuss the Diabetes Health Indicators Dataset (CDC, 2015), containing healthcare statistics and lifestyle survey information along with their diagnosis (or not) of diabetes, for the year 2015. Note that the dataset is biased, with 14% of individuals having diabetes. We start with a preprocessed version of the original questionnaire dataset, from which 21 features and the target variable are extracted. We select 10 out of 21 features to provide a more easily understandable problem for the reader.

We construct a causal graph (see Figure 9) relating these variables and train an SCM to model them, with which we finally compute their do-Shapley values. We design the graph using common sense. Note that any causal analysis depends on its graph being sound, but it can be replicated at any time once a better graph is found; for this reason, please take this as an illustrative example, since its conclusions regarding healthcare are not necessarily rigorous. We train a DCG with the same setup as before, except that every variable other than BMI is binary, so they are modelled with Bernoulli DCNs.

Our objective here is not to explain a ML model, but the data itself; particularly, how each variable affects the likelihood of diabetes. However, the effect of the noise is not as clear in a classifier as in a regressor, since $\phi_{E_X} = y - \mathbb{E}\left[Y \mid pa_Y\right] = y - P(Y \mid pa_Y)$; for an application of Theorem 4.8, please refer to Appendix F.2.

We compute do-SHAP for the first 1,000 test set entries, and measure FI. See Figure 10 a); HighBP, HighChol and BMI appear to be the most important variables, with Physical Activity, and fruit and vegetable intake having a less pronounced role.

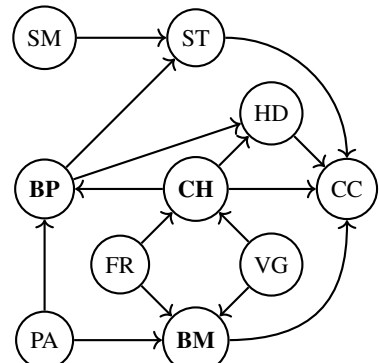

Figure 9: Diabetes Causal Graph, with variables Physical Activity (PA), Fruit (FR), Veggies (VG), Smoker (SM), BMI (**BM**), High Cholesterol (**CH**), High Blood Pressure (**BP**), Heart Disease or Attack (HD), Stroke (ST), Chol. Check (CC). Boldface letters denote $Pa_Y$, with $Y$, Diabetes, the target variable, not represented for clarity. We can skip modeling any non-ancestors of $Y$ (Proposition 4.2): SM, ST, HD and CC.

It is also important to consider the dependency between feature values and do-SHAP values; see the beeswarm plot in Figure 10 c), which shows clear-cut effects in do-SV sign and magnitude for HighBP and HighChol, with a more nuanced relationship between BMI (continuous) and do-SVs, which we plot in Figure 10 b); BMI 30 (typically categorized as obese) seems to be the cutting

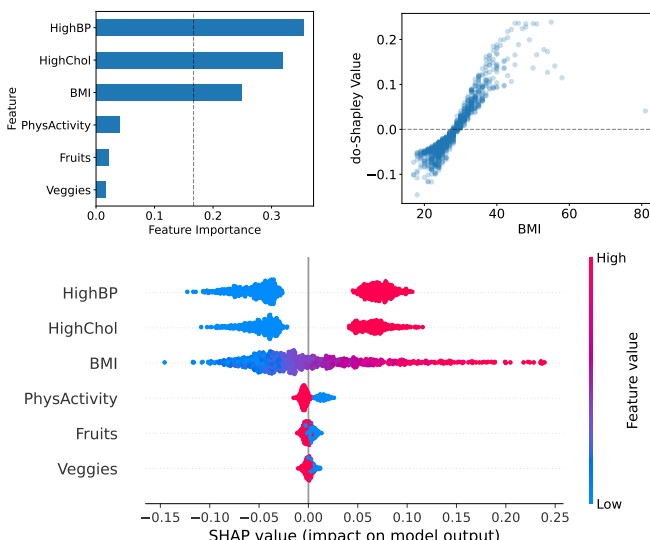

Figure 10: Diabetes Dataset. a) do-SHAP Feature Importance. Dashed vertical line represents uniform importance $(\frac{1}{K})$. b) Scatterplot between BMI value and do-SVs. c) do-SHAP beeswarm plot, relating do-SVs and feature values.

point after which higher BMI values increase the chances of diabetes up to 20%, while lower values decrease that likelihood up to 10%.

## F.2 BIKE RENTAL DATASET

We now study the Bike Rental Dataset (Fanaee-T & Gama, 2014), describing the number of rentals in a bike sharing service in Washington D.C., between 2011 to 2012 (both included), measured on an hourly basis, along with weather data and whether that day was a working day. Again, our objective in this case is to explain the data itself; particularly, how each variable affects the number of rented bikes.

We design a causal graph, depicted in Figure 11 (a). As stated before, please do not take the conclusions from this experiment as is, but merely as an illustrative example. We train a DCG with the same architecture and training parameters as the synthetic experiment, except that *hour* can be modeled with a uniform distribution on the integer interval $[0, 23]$. We train our DCG and compute the do-Shapley values for the test set entries. However, since we are operating on an inaccessible DGP in this case, we also want to measure the effect of the noise, employing Theorem 4.8.

We measure FI, represented in Figure 11 (b). Hour seems to be the major cause of the target variable, as is to be expected, followed by noise (given by the inherent variance of the target conditioned on its parents) and temperature, which also conditions on the likelihood of users renting bikes. Other variables also do have an impact, with only wind speed and weather (categorical, with three levels) having a less pronounced effect.

The relationship between feature values and do-SVs is more informative; we use scatter plots in order to study how each value affects the outcome; see Figure 12. *Hour* presents positive attribution during daytime from 8AM to 8PM, with night-time having negative attribution; *temperature*'s effect is mainly negative, with certain temperatures being less inviting for cycling (below 15ºC and above 35ºC); in the same way, humidity only affects past 80%, as well as wind speed, past 15 km/h.

## G COMPARISON BETWEEN DO-SVS AND NON-CAUSAL APPROACHES

In this appendix, we expand on the discussion about why do-SHAP results in more reliable explanations than its non-casual alternatives (marginal-SHAP and conditional-SHAP). We follow on the example presented in Section 1, here replicated in Figure 13 for reader's convenience.

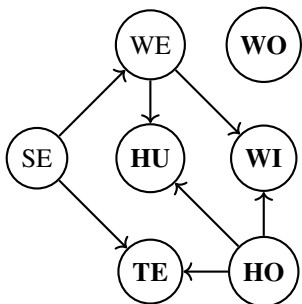 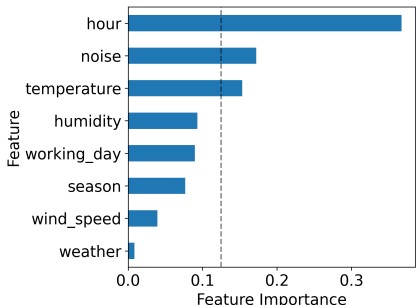

Figure 11: Bike Rental Dataset. a) Causal Graph with nodes Season (SE), Weather (WE), Humidity (HU), Temperature (TE), Working day (WO), Wind speed (WI), Hour (HO). Boldface letters denote $Pa_Y$, with target $Y$, Rentals, not represented for clarity. b) Feature Importance (FI) for each variable. Dashed vertical line represents uniform importance ($\frac{1}{K}$).

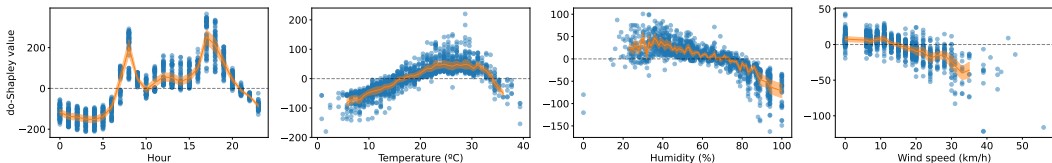

Figure 12: Bike Rental, continuous features' values against their SHAP values. Errors bars at 2-sigma.

Firstly, let us reason about how each method will behave in this particular Data Generating Process (DGP). In marginal-SHAP, consider for example $\nu(\{E\})$, where we would marginalize $A$ and $S$ regardless of the assigned value to $E$, thereby ignoring the impact that education level may have on the seniority level of the employee (their standing in the company), and producing out-of-distribution configurations $(a, e, s)$. Alternatively, with conditional SHAP, we would operate with $P(A, S \mid E = e)$, thereby including this dependency between $E$ and $S$, but also taking whichever value of $A$ would have generated this specific value $e$, which introduces in turn

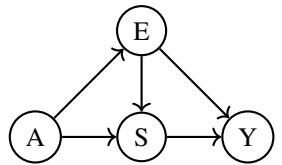

Figure 13: *Salary* causal graph.

an anti-causal effect ($E \to A$). Since both approaches ignore the causal structure, they incorporate non-causal effects that fail to reflect the real DGP, and would therefore lead to unreliable explanations. In contrast, do-SHAP does take into account this causal effect, by using the intervention $do(E = e)$, therefore affecting S ($E \to S$) while not affecting A ($E \leftarrow A$); not only that, but $A$'s effect is de-confounded by blocking the back-door path $E \leftarrow A \to S \to Y$.

Secondly, we will illustrate this reasoning with an example SCM corresponding to the same graph; see Equation (21). Let us consider Bernoulli r.v.s $A, E, S, Y$ with parameters $p_A, p_E, p_S, p_Y$, respectively. These parameters are computed following the aforementioned causal graph, with the following structural equations, which generate samples $(a, e, s, y)$ through Bernoulli sampling.

$$\begin{cases} p_A = 0.25 \\ p_E = 0.5a + 0.25 \\ p_S = 0.25a + 0.5e + 0.1 \\ p_Y = 0.5e + 0.3s + 0.1 \end{cases} \tag{21}$$

We evaluate marginal-SHAP, conditional-SHAP, and do-SHAP on this DGP. For that, we generate 1,000 background samples (used for marginalization in the first two methods) and 1,000 test samples to explain. We estimate marginal-SVs on test set samples $\mathbf{x}$ by approximating $\nu_{marg}(\mathbf{S}) := \mathbb{E}_{\mathbf{x}' \sim P(\mathbf{X})} \left[ P(Y \mid \mathbf{x_S}, \mathbf{x}'_{\mathbf{S}^c}) \right]$ through Monte Carlo with the i.i.d. background samples. Similarly, we estimate conditional-SVs by approximating $\nu_{cond}(\mathbf{S}) = \mathbb{E}_{\mathbf{x}' \sim P(\mathbf{X} \mid \mathbf{x_S})} \left[ P(Y \mid \mathbf{x_S}, \mathbf{x}'_{\mathbf{S}^c}) \right]$, averaging over those $\mathbf{x}'$ background samples that fulfill $\mathbf{x}'_{\mathbf{S}} = \mathbf{x_S}$. Finally, we compute do-SVs by es-

timating $\nu(\mathbf{S}) = \mathbb{E}_{\mathbf{x}' \sim P(\mathbf{X}|do(\mathbf{S}=\mathbf{x_S}))} \left[ P(Y \mid \mathbf{x_S}, \mathbf{x}'_{\mathbf{S}^c}) \right]$, sampling and intervening on the DGP directly with interventions $do(\mathbf{S} = \mathbf{x_S})$. We split the test samples for every factual combination $(a, e, s)$ and plot the corresponding estimations in the boxplots in Figure 14.

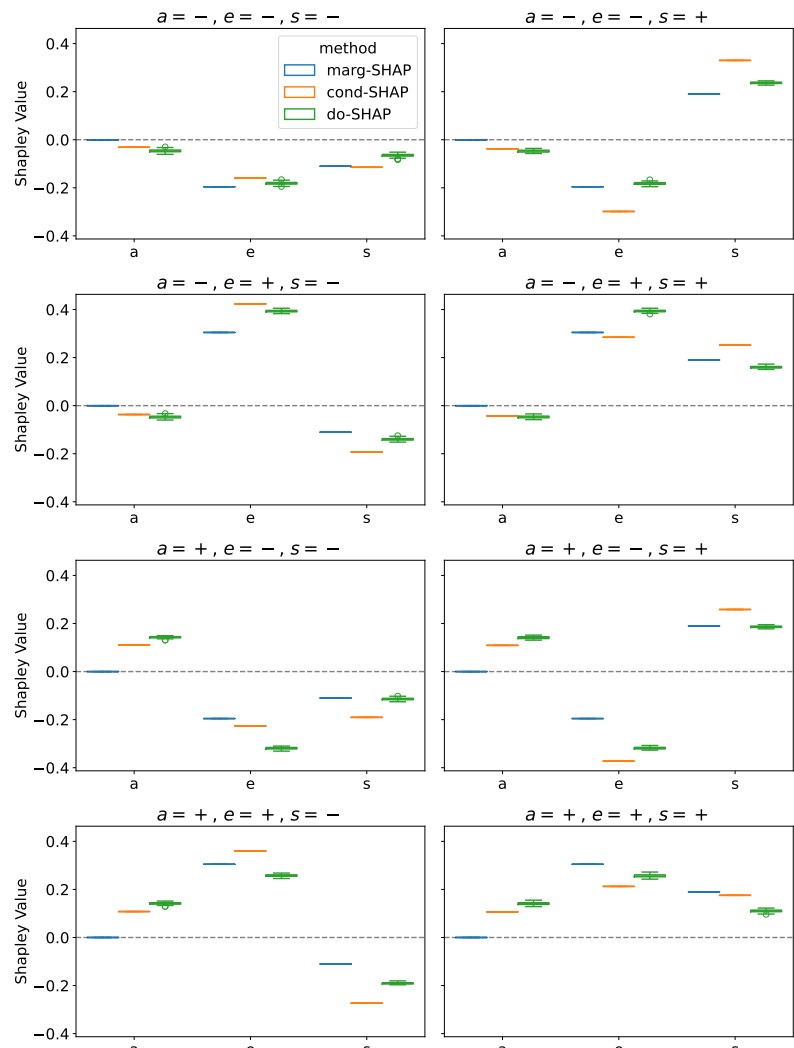

Figure 14: Salary example, comparison between marginal-SHAP, conditional-SHAP and do-SHAP for each input variable $(A, E, S)$ on every factual combination (title).

We use the results of this experiment to exemplify how marginal-SHAP and conditional-SHAP fail to address the behavior of the underlying DGP, hence providing explanations whose insights are not reliably applicable to the real world. Meanwhile, do-SHAP does take into account the corresponding data structure, and overcomes these weaknesses.

On the one hand, marginal-SHAP sets $\phi_A = 0$, since $A$ does not appear in $Y$'s structural equation; however, there is an effect from $A$ to $E$ and from $A$ to $S$, both of which in turn do have an effect on $Y$. As for $\phi_E$, it disregards the effect of $E$ on $S$ while also confounding the back-door effect $E \leftarrow A \rightarrow S \rightarrow Y$. Finally, for $\phi_S$, it is closer to an intervention with do-SHAP, since $S$ does not affect any variable other than $Y$, but it is unable to control the back-door effect ($S \leftarrow E \rightarrow Y$ or $S \leftarrow A \rightarrow E \rightarrow Y$), which explains the difference with do-SHAP.

On the other hand, conditional-SHAP results in similar attributions for $\phi_A$ to do-SHAP, due to the fact that $\{A\}, \{A, E\}$ and $\{A, E, S\}$ have identical $\nu(.)$ values for conditional-SHAP and do-SHAP. However, $\{A, S\}$ introduces an anti-causal flow when conditioning ($S \rightarrow E$), which explains the

difference with do-SHAP. This is more apparent in $\phi_E$ and $\phi_S$, where the anti-causal flow inevitably affects the estimations.

In summary, both marginal-SHAP and conditional-SHAP misrepresent the underlying DGP, given that they ignore its underlying causal structure. This is the reason why do-SHAP results in more reliable explanations than the alternatives.

## H    DO-SHAP ESTIMATION EXAMPLE

In this appendix, we explain how to employ the estimand-agnostic approach in practice to estimate do-SVs. This is meant as an illustrative written explanation; for a code example, please refer to the experiments code, which will be submitted along with the camera-ready version of this work.

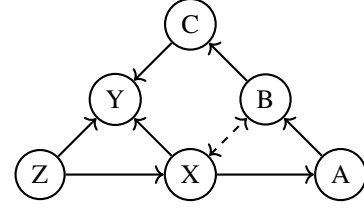

We will focus on the graph introduced in Section 5.1, here replicated in Figure 15 for reader's convenience. We will employ the semi-Markovian version of this example (meaning, there is a latent confounder $U_{\{X,B\}}$ between $X$ and $B$). We will assume the (measured) data distribution is composed of unconstrained continuous r.v.s, with an unknown prior distribution for $U_{\{X,B\}}$, but with a known causal graph $\mathcal{G}$.

Figure 15: Synthetic semi-Markovian graph.

In terms of notation, let $Pa'_X = Pa_X \sqcup \mathcal{U}_{\{X,\cdot\}}$ be the concatenation of the parents of a certain node $X \in \mathbf{X}$ and any latent confounders pointing to it. For example, in our current graph, $Pa'_B = (A, U_{\{X,B\}})$.

### H.1    DO-SHAP IDENTIFIABILITY

Before starting, we need to confirm that do-SVs are indeed identifiable in this particular graph; otherwise, two SCMs trained on the same distribution may output different do-SV estimations, rendering them useless.

If there was a graphical criterion that fit the structure of our graph $\mathcal{G}$, it could be used at this step; for instance, if there are no latent confounders in a graph, do-SHAP is trivially identifiable. Since this is not the case, we need to test it using the ID algorithm (Shpitser & Pearl, 2006a). See (Tikka & Karvanen, 2017) or (Pedemonte et al., 2021) for implementations in R and Python, respectively.

To guarantee do-SHAP identifiability, we need to test it for each coalition $\mathbf{S}$. In other words, ensure that $\nu_{\mathbf{x}}(\mathbf{S}) := \mathbb{E}[Y \mid do(\mathbf{S} = \mathbf{s})]$ is identifiable $\forall \mathbf{S} \subseteq \mathbf{X}$. Therefore, we run the ID algorithm on each of the $2^5 = 32$ queries $\nu(\mathbf{S})$; if all are identifiable, so are the do-SVs. Note that this test is independent to the data distribution, as it only requires the corresponding graph structure $\mathcal{G}$.

Given that this is a small graph, with only 5 $\mathbf{X}$ variables, amounting to 32 coalitions, it is feasible to test for identifiability before starting the process. In the general case, with potentially bigger graphs and $2^{|\mathbf{X}|}$ coalitions to test, this becomes infeasible or very expensive, and it is therefore recommended to test for identifiability *during* do-SV estimation; we will indicate where in the process this is taken care of in the following subsections.

### H.2    SCM IMPLEMENTATION

Given that $\mathcal{G}$ contains a latent confounder, we choose DCGs (Parafita & Vitrià, 2022) as the SCM architecture, and for generality (to adjust to more complex data distributions) employ Normalizing Causal Flows (NCFs) as the node architecture. Refer to Appendix E.1.1 for a possible implementation of NCFs (without domain adjustment, given that our variables are unconstrained real-valued r.v.s). These NCFs provide a general function $X = f_X(E_X, \Theta_X(Pa'_X))$, with $E_X$ the corresponding exogenous noise signal for node $X$, which is used to sample new values $x \sim P(X \mid Pa'_X)$ as well as to compute the log-likelihood of these values given its parents: $P(x \mid pa'_X)$. The choice of prior distribution for the exogenous signal $E_X$ and for the latent confounder $U_{\{X,B\}}$ is irrelevant, as long as the desired queries to estimate are identifiable (which has been previously tested) and

the particular choice of prior guarantees enough modeling capacity to represent the data distribution $P(\mathbf{X})$. In this particular case, we choose a $\mathcal{N}(0, 1)$ for both priors.

Regarding the function $\Theta_X(Pa'_X)$, it returns the appropriate function parameters $\theta_X$ for each node's function $f_X$. These parameters define the shape of the distribution $P(X \mid Pa'_X)$ and as such depend on the values $pa'_X$ of $Pa'_X$. After that, $f_X$ only depends on the parameters $\theta_X$ and the exogenous noise $\varepsilon_X \sim P(E_X)$. This function $\Theta_X(.)$ could be modeled with a simple Multilayer Perceptron (MLP) node by node (one for each node), but this would inevitably result in overfitting for larger graphs. Instead, we employ the Graphical Conditioner presented in (Parafita & Vitrià, 2022), a single MLP network that takes every node $\mathbf{X}$ and latent confounders $\mathcal{U}$ as the input and returns all parameters $\{\theta_X \mid X \in \mathbf{X}\}$ as the output. By using a particular training and inference strategy, the Graphical Conditioner allows to compute each of these functions $\Theta_X$ independently through a single network, thereby reducing training time and overfitting risk.

## H.3  SCM TRAINING

DCGs are trained with Maximum Likelihood Estimation, so we will define, for a particular sample $\mathbf{x} \sim P(\mathbf{X})$, the Negative Log-Likelihood (NLL) loss to minimize as the training objective. Given the graph structure $\mathcal{G}$ of the data distribution to model, we can derive two formulas, one for the Markovian case (no latent confounders),

$$\mathcal{L} := -\log P(\mathbf{x}) = -\sum_{X \in \mathbf{X}} \log P(x \mid pa_X), \tag{22}$$

the other for the semi-Markovian case (with latent confounders),

$$\mathcal{L} := -\log P(\mathbf{x}) = -\mathbb{E}_{\mathcal{U}} \left[ \exp \sum_{X \in \mathbf{X}} \log P(x \mid pa'_X) \right]. \tag{23}$$

In our example, we need to employ Equation (23), given that $\mathcal{U} = \{U_{\{X,B\}}\} \neq \varnothing$. This latter expectation can be estimated by generating $M$ i.i.d. samples from $P(\mathcal{U})$ and averaging the results of the expectation's contents. The terms $\log P(x \mid pa'_X)$ can be estimated by the predefined node architectures (NCF in our example) using a simple function. Finally, the Monte Carlo average can be computed using the log-sum-exp trick for numerical stability.

By running an optimization algorithm (e.g., AdamW (Loshchilov & Hutter, 2019)) on the average of these NLL losses for random mini-batches of samples, we can learn the data distribution $P(\mathbf{X})$ with our SCM, from which we can now estimate (identifiable) do-SVs.

## H.4  DO-SHAP ESTIMATION

We cover this last step in three parts.

Firstly, we need to run the SHAP formula. In this case, with only 5 variables, we could employ the exact formula directly, using Equation (1). Instead, we exemplify the more general approach, compatible with larger graphs, with the permutations formula in Equation (2), which we approximate via Monte Carlo as described in Section 3.4, here re-established for reader's convenience:

$$\phi_{\nu_{\mathbf{x}}}(X) = \mathbb{E}_{\pi \sim \mathcal{U}(\Pi(\mathbf{X}))} \left[ \nu_{\mathbf{x}}(\mathbf{X}_{\leq_\pi X}) - \nu_{\mathbf{x}}(\mathbf{X}_{<_\pi X}) \right], \tag{24}$$

where the expectation over permutations $\pi$ is estimated by generating $M$ i.i.d. permutations uniformly, and the internal $\nu_{\mathbf{x}}(.)$ terms are estimated as described in the following. However, before we run the $\nu_{\mathbf{x}}$-estimation procedure, we employ FRA to compute the Frontier-Irreducible subsets $\mathbf{S}' \subseteq \mathbf{S}$ and, as such, reduce the number of $\nu_{\mathbf{x}}$-computations required, by using a cache.

Secondly, let us discuss FRA. Consider the permutation $\pi = (A, B, Z, X, C)$. We need to attempt to reduce sets $\mathbf{X}_{\leq_\pi X} = \{A, B, Z, X\}$ and $\mathbf{X}_{<_\pi X} = \{A, B, Z\}$. By running the FRA algorithm (see Algorithm 2 for a simpler application with sets), we can see that both sets can be reduced by removing $A$, since $B$ acts as a frontier between $A$ and $Y$. Hence, we compute both values $\nu_{\mathbf{x}}(\{B, Z, X\})$

and $\nu_{\mathbf{x}}(\{B, Z\})$ and store their results in a cache for later look-up. If, on another permutation $\pi'$, we encounter coalitions with irreducible sets that have been computed before, we can retrieve the results from the cache directly, thereby reducing the number of required computations.

At this point, if we have not tested identifiability at the beginning of the process, we must confirm identifiability of each encountered $\nu(\mathbf{S}')$ query (with the irreducible set) before proceeding. If the query is deemed unidentifiable, the process must halt with an error, as do-SVs cannot be estimated in this particular graph.

Thirdly, let us describe how to estimate an arbitrary coalition value, $\nu_{\mathbf{x}}(\mathbf{S})$, which is accomplished with a general sampling procedure. In order to estimate this query, we need to generate $M$ i.i.d. samples $\mathbf{x}^{(i)} \sim P(\mathbf{X} \mid do(\mathbf{S} = \mathbf{s}))$. We start by generating values $\varepsilon^{(i)} \sim P(\mathcal{E})$ and $u^{(i)} \sim P(\mathcal{U})$ from their respective prior distributions. Afterwards, we go node by node $X \in \mathbf{X}$ following any topological order of the graph, using the corresponding sampling functions $x^{(i)} = f_X(\varepsilon_X^{(i)}, \Theta_X(pa_X'^{(i)}))$ unless the variable $X$ is in the intervened coalition $\mathbf{S}$, in which case $x^{(i)}$ becomes the corresponding value from the to-be-explained sample $\mathbf{x}$. After we have sampled from every variable in the graph, we have a joint sample $\mathbf{x}^{(i)} \sim P(\mathbf{X} \mid do(\mathbf{S} = \mathbf{s}))$. We pass each of these samples through our model $f_Y(.)$ to compute the corresponding $y^{(i)}$ values, which will finally be averaged for the final estimation of $\nu_{\mathbf{x}}(\mathbf{S})$.

Finally, we can add further optimizations to these procedures. Consider the tuple $\nu_{\mathbf{x}}(\pi) = (\nu_{\mathbf{x}}(\pi_{:k}))_{k=0..K}$, with $\pi_{:k}$ the set of variables up to index $k$ on $\pi$ (inclusive) (note that $\pi_{:0} = \varnothing$). When computing SVs, instead of using the formula directly, we sample permutations $\pi$, compute the corresponding tuples $\nu_{\mathbf{x}}(\pi)$ and, from there, their diff-tuple $\Delta\nu_{\mathbf{x}}(\pi) = (\nu_{\mathbf{x}}(\pi_{:k}) - \nu_{\mathbf{x}}(\pi_{:k-1}))_{k=1..K}$. Note that each of these $\Delta\nu_{\mathbf{x}}(\pi)_k$ terms is the difference in the SHAP formula for variable $X := \pi_k$, so we can update our do-SV estimations $(\phi_X)_{X \in \mathbf{X}}$ simultaneously, which accelerates this computation by reducing the number of FRA-cache accesses and employing tensor operations. Along with this, other estimation approaches, such as antithetic sampling for these permutations, can be used to obtain better estimator efficiency, hence further reducing the number of required permutations.

## H.5 Final considerations

Putting all of this together amounts to a seemingly complex method to estimate feature attribution: from finding the assumed Causal Graph $\mathcal{G}$, defining the appropriate SCM architecture, training such a model, estimating the $\nu$ values, running FRA to avoid computations, to finally arriving at the do-SV estimations. However, given already-implemented SCM architectures with appropriate training and estimation procedures ready for use, estimand-agnostic do-SHAP is made practical. For this reason, we advocate for an open-source library bringing together different SCM architectures for easier switching between approaches, facilitating the general applicability of estimand-agnostic methods, and as a result, of do-SHAP explanations.

## I Impact statement

The goal and byproduct of this work is a tool for the estimation of do-SVs on an arbitrary black-box system we wish to explain, natural or artificial. As an explainability tool, the core impact of this research is positive, as it will push towards several desirable goals: on top of being a useful tool both for science and business, as it allows to understand underlying systems in the data and apply that knowledge with informed decisions, explanations answer to several concerns about the trustworthiness of AI systems; e.g., the *right to explanation* on human-facing decision systems, debugging complex black-box systems, and better transparency and accountability. Additionally, since it constitutes a tool for auditing these systems, explainability paves the way for AI regulation, necessary to protect human rights against the blind application of powerful but opaque AI systems.

One potential negative application of these techniques is in terms of willingly or unwillingly obfuscating harmful behaviour in black-box models. Given the complexity of these techniques and the subsequent analysis required to derive conclusions from its outputs, explainability techniques could be used to provide a superficial layer of supervision and result in misleading conclusions about the behaviour of the system. Great care with respect to the assumptions and outputs of these tools must be taken in their application.

