# OpenReview forum: "Scalable do-Shapley Explanations with Estimand-Agnostic Causal Inference"
_ICLR.cc/2025/Conference — Submitted to ICLR 2025_

### Official Review · Reviewer_DDJF · 2024-10-31

**Soundness:** 2
**Presentation:** 3
**Contribution:** 2
**Rating:** 6
**Confidence:** 4

**Summary:**

In this work, the authors present a method for efficient estimand-agnostic computation of causal do-SHAP values, i.e., explanations of models with causal interventions. To this end, the authors present the Frontier-Reducibility Algorithm (FRA), a novel algorithm to decide a cacheing scheme for causal models. FRA achieves this speedup by avoiding unnecessary re-computation of parts of the graph that are independent of the causal effects. Furthermore, explainability of inaccessible data generating processes is considered.

**Strengths:**

1. Overall, the mathematical presentation of the contributions is consistent and clear.

2. The FRA algorithm is shown to setup an efficient cache for speeding up the runtime of the causal do-SHAP explanation method (Fig. 4 c).

3. The FRA algorithm is shown to have a minor impact on overall computation times (Fig. 4 b).

4. The presented method is shown to be applied in cases where the data generating process is not available (although the causal structure is assumed to be known).

**Weaknesses:**

1. Although Figure 2 compares a range of models, only Deep Causal Graphs (DCG) are considered in the second experiment in Figure 4. As CNF performs better in Figure 2 (higher log-likelihood, lower estimation loss), even if DCG is supposedly faster (which is not shown in the experiments), it would be interesting to see if, e.g., FRA may enable the use of the more accurate but on its own slower model.

2. The abstract may misrepresent the contributions of the paper. First, it states that they "propose estimand-agnostic Causal Inference", which is (as outlined on page 2) prior work. Stating that estimand-agnostic method are employed in this paper would be more clear in this regard. Second, it states that the presented method is "making do-SHAP feasible on arbitrarily complex graphs". Although a significant speedup is shown (Fig. 4 c), the exponential growth in computation time persists.

3. As stated in the abstract, part of the motivating hypothesis for this paper is that the "reliance on estimands hinders scalability" (abstract). But, a comparison with such methods is missing in the evaluation.

Minor:

1. The abbreviation DCG is only introduced in the appendix (page 24).

2. A confusing double negation in the footnote on page 9 ("nor have not").

**Questions:**

1. Why is FRA's execution time in Figure 4 (b) going down with a growing number of variables and only starting to grow once about 8 variables are involved?

2. In Figure 4 (c), three variants are compared (no cache, cache, and FRA cache). It did not become apparent to me how the non-FRA cache is setup (line 504 simply states "employ a cache").

---

> ### Author Response · Authors · 2024-11-14
> **Rebuttal**
>
> We thank the reviewer for their review. We will address each of the concerns in the following:
>
> W1: The reason for choosing DCGs over CNFs was, mainly, because they admit latent confounders (provided identifiability) while CNFs do not. Even if experiment 5.2 did not have latent confounders, for consistency with the experiment in Appendix E.1.2 (the semi-Markovian case), and applications in Appendices F.1 and F.2 (the two real-world datasets), all including latent confounders, it was preferable to always employ the same model for ease of readers wanting to reproduce the code (which will be included with the camera-ready version). More importantly, estimation time was a crucial consideration for experiment 5.2, given the number of replications and the increasing number of features. As stated, CNFs were orders of magnitude slower than DCGs during estimation, which would result in a prohibitively expensive experiment time-wise in comparison to DCGs, more so when the choice of model was not important in this particular experiment (longer SCM estimation times would simply multiply by the number of $\nu$ computations, which is reduced by FRA). We deliberately chose a fast SCM so as to compare estimation time ($\nu$ evaluation) against the FRA algorithm (the tests required to identify irreducible sets); since FRA is negligible time-wise compared to a faster SCM, it is only advantageous to apply it for slower SCMs. Besides, the objective in this experiment was to compare between no-cache, cache and FRA-cache, and therefore the choice of model was not particularly relevant. Indeed, as you say, FRA may make more powerful SCM architectures like CNFs feasible for do-SHAP estimation, but there was no need to try one such architecture for this demonstration.
>
> W2: Thank you for pointing this out; the choice of words in the abstract may misrepresent our contribution (which is better stated in the main text, see line 60). The intended meaning was that we propose the estimand-agnostic approach *for* do-SHAP, which is a significant contribution given that the original do-SHAP paper did not contemplate such a possibility, as stated by the authors (see line 58), proving it impractical. By identifying this connection and proving its feasibility and accuracy in estimating these attributions through our experiments, we can make do-SHAP feasible for arbitrary graphs, which, in our opinion, is an important contribution to the literature, especially since it could help promote the use of causality-aware attribution methods.
>
> Regarding your second comment, when we mention that we make do-SHAP feasible for arbitrarily complex graphs, we refer to the fact that, if we employed the estimand-based approach, it would require to manually employ estimation techniques for $2^{|X|}$ estimands, one for each query $\nu(S)$, which is simply infeasible in practice. Instead, by training one single model and employing general procedures for interventional queries like these $\nu(S)$, the estimand-agnostic approach can *automatically* adapt to any kind of graph. It is true that the exponential nature of SHAP remains, which is why it is important to employ the approximation approach (section 3.4) and our FRA algorithm and the derived properties (section 4.2) to further reduce computations. We do not claim to avoid the exponential complexity of SHAP, but we do provide a way to compute these terms feasibly in practice (estimand-agnostic) and more efficiently (FRA), as proven in section 5.2.
>
> Communicating these nuances has proven to be hard, but we will rectify the specific wording of the abstract to avoid misrepresenting our contribution. Thank you for pointing this out.
>
> (Rebuttal continues in the following message)

---

> > ### Author Response · Authors · 2024-11-14
> > **Rebuttal (part 2)**
> >
> > W3: This is related to the previous point. Scalability in this case cannot be measured empirically, since it refers to the fact that the estimand-based approach requires the manual definition of each estimand formula and employing techniques to adapt to that particular probabilistic formula. Given a graph with 5 variables, there are 32 such estimand formulas that the user needs to follow manually by training/employing probabilistic models, possibly many of them. Instead, we propose employing a single SCM that can answer any of these queries automatically, using a single general procedure that adapts to these queries. That is the power of the estimand-agnostic approach, and why it makes it feasible to compute do-SHAP on arbitrarily complex graphs (the exponential nature of SHAP notwithstanding). Therefore, it would not make sense to test the estimation time of training a DCG and computing these do-SVs with estimand-agnostic procedures in contrast with the manual evaluation of each $\nu$ term individually that the estimand-based approach requires.
> >
> > MinorW1: Thank you for pointing this out. It was meant to be introduced in line 442 but it escaped our notice. We will rectify in the camera-ready version.
> >
> > MinorW2: Thank you, we will rectify this point.
> >
> > Q1: A possible explanation could be because $\mathcal{G}_{K, p}$, the class of graphs from which we draw in this experiment, requires all nodes to be ancestors of Y and not all nodes can be parents of Y (see the paragraph starting on line 464). With these two restrictions, $\forall i,$ either $X_i \rightarrow Y$ or $\exists j > i$ s.t. $X_i \rightarrow X_j$, which in low-K graphs restricts the number of possibilities considerably, biasing towards denser graphs, regardless if $p$ is initially lower (because we employ rejection sampling). This particular structure may induce more tests than when $K$ is higher. Regardless, the trend is steady and apparent past $K>7$, which is why we did not give it more importance.
> >
> > Q2: This experiment uses the approximation formula with permutations, so for any permutation, we may encounter a certain subset $\nu(S)$ repeatedly. A non-FRA cache would cache the result once and thereafter avoid any following computations for that specific coalition. The FRA cache, instead, first reduces the coalition to the irreducible set $S’$ and then caches its $\nu(S’)$; thereafter, if the same coalition $S$ appears, or a new coalition $S’’$ appears resulting in the same irreducible set $S’$, its result will already be precomputed. We will clarify this point in the paper, thank you for pointing it out.

---

> > > ### Comment · Reviewer_DDJF · 2024-11-24
> > > **Response to the rebuttal**
> > >
> > > Thank you for the response. I think instead of promising that things will be done, the authors could have uploaded a revised version of the paper. This would have been far more tangible to judge.
> > >
> > > I retain my rating.

---

> > > > ### Author Response · Authors · 2024-11-27
> > > >
> > > > We hope you had the opportunity to review our global comment outlining the updates in the revised version of the paper. We have thoroughly addressed all your concerns, along with those raised by other reviewers. We trust these revisions, supported by our detailed rebuttals, effectively resolve the issues you highlighted. We kindly ask you to reconsider your score in light of these improvements to support the publication of our work. Thank you for your time and consideration.

---

### Official Review · Reviewer_vEEg · 2024-11-03

**Soundness:** 2
**Presentation:** 2
**Contribution:** 2
**Rating:** 5
**Confidence:** 2

**Summary:**

This paper proposes Frontier Reducibility Algorithm to make the computation of do-shapley values more efficient. Specifically, the authors propose to use estimand-agnostic approach for calculating each combinatorial terms in do-shapley and simplified the process based on graphical information, e.g., by omitting non-ancestors. The authors also empirically validates the proposed method by synthetic data.

**Strengths:**

1. The topic is interesting and important to many related fields.
2. The proposed Frontier Reducibility Algorithm seems to be more computationally efficient than baselines.

**Weaknesses:**

1. The presentation can be improved. The notation is not aligned, e.g., $\mathbf{V}$ refers to set of variables (line 117), while later $\mathcal{V}$ is used (line 125).

2. I am not sure about whether the contribution is enough. The first claimed contribution is the use of estimand-agnostic approach, which is not novel (from Parafita&Vitria 2022). The second claimed contribution is the Frontier Reducibility Algorithm for calculation of do-shapley based on Thm 4.7, and yet it is related to Lutheretal. (2023). The third claimed contribution is a novel explainability strategy with do-SVs,
which also looks non-significant. Due to that I am not very familiar with this particular field I will defer to other reviewers regarding this point.

**Questions:**

Could you provide a more detailed description on the example in Figure1? specifically, in terms of why do-shapley would be better than e.g., marginal or conditional shapley? It would also help if the authors can use the example in Figure 1 to explain the method in Section 4.3.

---

> ### Author Response · Authors · 2024-11-14
> **Rebuttal**
>
> We thank the reviewer for their review. We will address each of the concerns in the following:
>
> W1. This specific instance is not a mistake, since both terms are not the same. The first, the set of nodes $\textbf{V}$ in the causal graph, consists of the SCM’s measured variables $\mathcal{V}$, their corresponding exogenous noise variables $\mathcal{E}$ and (if any) the latent confounders $\mathcal{U}$. $\textbf{V}$ includes all random variables in the SCM (not only $\mathcal{V}$), despite not all of them showing explicitly in the causal graph representation, as discussed in footnote 2, hence why the notation is different. We were very particular about the use of notation, so any perceived inconsistencies may not be so, but there may have been some errors in writing that escaped our notice. If you identified any others, we would greatly appreciate you notifying us so we can rectify them.
>
> W2. We will address this concern in detail, since it seems to be the most pressing point.
>
> Regarding the first contribution, while this work does not propose the estimand-agnostic framework in itself (Parafita & Vitrià 2022), it does: 1) make do-SHAP feasible (since it was impractical due to its reliance on estimands) by employing the estimand-agnostic approach, and 2) prove that SCMs can indeed estimate do-SVs effectively, despite not employing estimands like the original authors proposed.
>
> The FRA algorithm contribution has many facets. Firstly, it is a novel set of properties employing the rules of do-calculus to derive these irreducible sets. Note that this approach is completely different from the work in (Luther et al., 2023), since theirs was exclusively conditional and did not consider the causal structure in the data. We mention Luther’s work because of its connection in employing conditional independence to identify instances where $\nu(S \cup \{k\}) = \nu(S)$, but we go further by considering: 1) the causal framework, where these $\nu$ entail causal queries with $do(.)$ interventions, and 2) by identifying the irreducible set $S’$ such that $\nu(S) = \nu(S’)$, thereby not reducing by one variable, but potentially many, resulting in more computations avoided since they all collapse on the same irreducible set, already cached. Secondly, we derive an algorithm to efficiently identify these irreducible sets, which is vital in the context of SHAP given its exponential complexity (albeit alleviated by the approximated approach, discussed in section 3.4).
>
> Finally, the third contribution, theorem 4.8, may seem a simple, intuitive result, but requires an extensive demonstration employing the rules of do-calculus (see Appendix D.3). Thanks to this proof, we can now explain these inaccessible DGPs in a human-understandable manner, since the SHAP values now add up to the actual value for $Y$, whereas before it simply added to $\mathbb{E}[Y \mid \widehat{\textbf{x}}]$, which need not be close to the actual, factual value of $Y$, proving confusing for the end-user when receiving these explanations. Intuitively, we might have attributed this difference on the exogenous noise signal; our theorem, however, proves this fact (under certain assumptions). That is the importance of the claim.
>
> Finally, all of these contributions are tested with our experiments and showcased within two real-world datasets, in section 5 and Appendixes E-F.
>
> (Rebuttal continues in the following message)

---

> > ### Author Response · Authors · 2024-11-14
> > **Rebuttal (part 2)**
> >
> > Q1: Reviewer #98yc had a similar question. Let me address it here again for your convenience.
> >
> > By focusing on a simple example (Figure 1), we can see how the two non-causal alternative definitions to the Shapley game fail to address the behavior of the underlying mechanism. In marginal SHAP, consider for example $\nu(\{E\})$, we would marginalize A and S regardless of the assigned value to E, thereby ignoring the impact that education level may have on the seniority level of the employee (their standing in the company), and producing out-of-distribution configurations (a, e, s). Therefore, it would ignore an important mechanism in the data. Alternatively, with conditional SHAP, we would operate with $P(A, S \mid E=e)$, thereby including this dependency between E and S, but also taking whichever value of A would have generated this specific value e, introducing in turn an anti-causal effect from E to A, which we know is not possible. Both approaches, because they ignore the causal structure, incorporate non-causal effects that fail to reflect the real-world data generating process, and would therefore lead to unreliable explanations. In contrast, do-SHAP does take into account this causal effect, by using the intervention $do(E=e)$, therefore affecting S (E → S) while not affecting A (E ← A). This is why, only by considering the causal structure, and therefore employing do-SHAP rather than the non-causal alternatives, can we obtain reliable explanations that will translate to the real world if we applied these interventions.
> >
> > This example is explained very briefly in the paper because of space concerns. We will extend this explanation for further clarity in the camera-ready version.
> >
> > Regarding how to apply our method to this example, it would be as follows. Since this is only a 3-variable graph (plus Y), we will employ the exact formula for SHAP (eq. (2)) for illustrative purposes. For bigger graphs, the approximated method would be preferred.
> >
> > Consider the do-SV for variable E; any permutation of the three variables, for example, $\pi = (A, E, S)$ results in two sets, one including E and its predecessors, $\textbf{S} = \{A, E\}$, the other excluding E, $\textbf{S} = \{A\}$. We then compute $\nu$ on these subsets $\textbf{S}$ and substract their results, by employing the trained SCM on the causal query $\mathbb{E} [f_Y(A, E, S) \mid do(\textbf{S}=\textbf{s})]$, which can be estimated by applying, for example, the procedures detailed in (Parafita & Vitrià, 2022). We follow the SHAP formula for all other permutations and finally arrive at the desired do-SV. Regarding how to define and train this SCM, it depends on the actual implementation of the chosen SCM architecture; see also (Parafita & Vitrià, 2022) for an example with DCGs and their framework’s source code.
> >
> > We hope that this example helped in understanding how the estimand-agnostic approach is applied to do-SHAP. We will include an extended version as an additional appendix on the camera-ready version, to help readers understand the proposed method.

---

> > > ### Comment · Reviewer_vEEg · 2024-11-26
> > >
> > > Thank you very much for the response. It would help if the authors could upload the revised version for us to see the update. Currently I will keep my rating unchanged.

---

> > > > ### Author Response · Authors · 2024-11-27
> > > >
> > > > We hope you had the opportunity to review our global comment outlining the updates in the revised version of the paper. We have thoroughly addressed all your concerns, along with those raised by other reviewers. We trust these revisions, supported by our detailed rebuttals, effectively resolve the issues you highlighted. We kindly ask you to reconsider your score in light of these improvements to support the publication of our work. Thank you for your time and consideration.

---

### Official Review · Reviewer_98yc · 2024-11-06

**Soundness:** 3
**Presentation:** 1
**Contribution:** 2
**Rating:** 5
**Confidence:** 3

**Summary:**

The paper proposes two key contributions: first, it introduces an estimand-agnostic approach based on SCMs to compute causal effects with a single model, and hence improving scalability. Second, it presents a Frontier-Reducibility Algorithm to enhance do-SHAP's efficiency. The authors validate their method through empirical tests to demonstrate the effectiveness of their approach in generating explanations for ML models.

**Strengths:**

The paper identifies and addresses the limitations of traditional SHAP, which do not adequately consider causal relationships in data. By focusing on causal structures, it enhances the reliability of model explanations.

**Weaknesses:**

1/ The paper is poorly written and may not meet the required standards. It fails to highlight its real-life applications. In particular, the introduction does not convince readers of the need to incorporate causal structure into Shapley.

2/ For the do-SV (defined in Eq. 3) to be identifiable, I believe that specific assumptions are needed. However, they are not clearly stated in the paper. Do we need standard assumptions such as SUTVA, consistency, etc? Is it assumed that the confounders are all observed?

3/ Experiments focus on estimation error of do-SHAP and the speedup of FRA, but not on why the proposed method is better than SHAP.

4/ It would be valuable to compare the proposed method with existing explainable method such as:

Ribeiro, M. T., Singh, S., & Guestrin, C. (2016). "Why should I trust you?" Explaining the predictions of any classifier. Proceedings of the 22nd ACM SIGKDD International Conference on Knowledge Discovery and Data Mining.

Lundberg, S. M., & Lee, S. I. (2017). A unified approach to interpreting model predictions. Advances in Neural Information Processing Systems.

Fisher, P. (2018). "Permutation feature importance." In Machine Learning: A Probabilistic Perspective by Kevin P. Murphy.

Ribeiro, M. T., Singh, S., & Guestrin, C. (2018). Anchors: High-Precision Model-Agnostic Explanations. AAAI Conference on Artificial Intelligence.

Selvaraju, R. R., Cogswell, M., Das, A., et al. (2017). Grad-CAM: Visual Explanations from Deep Networks via Gradient-based Localization. International Conference on Computer Vision.

Although these methods do not consider causal structure, it is valuable to compare with them to showcase superiority of the proposed method in generating explanation.

5/ This is a minor comment: It is unnecessary to include so much bold text and highlighting in the paper.

Overall, I believe that the presentation of the paper needs to be improved, and more experimental evaluation is necessary.

**Questions:**

Please refer to Weaknesses.

---

> ### Author Response · Authors · 2024-11-14
> **Rebuttal**
>
> We thank the reviewer for their review. We will address each of the concerns in the following:
>
> W1: We would be grateful if the reviewer were more concrete regarding the claim that the paper is poorly written and not meeting the required standards. We are willing to improve upon the text and, apart from the bold text issue mentioned in W5, no further measures are mentioned on how to address this complaint.
>
> Regarding the need to incorporate causal structure into Shapley, we do address it, albeit briefly, in the paragraph starting on line 41. By focusing on a simple example (Figure 1), we can see how the two non-causal alternative definitions to the Shapley game fail to address the behavior of the underlying mechanism. In marginal SHAP, consider for example $\nu(\{E\})$, we would marginalize A and S regardless of the assigned value to E, thereby ignoring the impact that education level may have on the seniority level of the employee (their standing in the company), and producing out-of-distribution configurations (a, e, s). Therefore, it would ignore an important mechanism in the data. Alternatively, with conditional SHAP, we would operate with $P(A, S \mid E=e)$, thereby including this dependency between E and S, but also taking whichever value of A would have generated this specific value e, introducing in turn an anti-causal effect from E to A, which we know is not possible. Both approaches, because they ignore the causal structure, incorporate non-causal effects that fail to reflect the real-world data generating process, and would therefore lead to unreliable explanations. In contrast, do-SHAP does take into account this causal effect, by using the intervention do(E=e), therefore affecting S (E → S) while not affecting A (E ← A). This is why, only by considering the causal structure, and therefore employing do-SHAP rather than the non-causal alternatives, we can obtain reliable explanations that will translate to the real world if we applied these interventions.
>
> This example is explained very briefly in the paper because of space concerns. However, we will extend this explanation along these lines for the camera-ready version.
>
> On the other hand, on the topic of real-world applications, we did include two real-world examples in appendix F; although reviewers are not asked to read the appendix, we do include them and both are mentioned in the paper, in line 412.
>
> W2: The required assumptions are stated in the paper. Note that we are not operating from the Potential Outcomes perspective, where the assumptions you mentioned (SUTVA, consistency) are standard and always mentioned, but from the Causal Graph perspective, where the assumed causal graph G itself encodes the required assumptions. Our main assumption is that the graph is known and from there we can infer whatever assumptions are necessary. See Assumption 4.1 (line 245). “We assume $P(\mathcal{V})$ to be strictly positive,” (positivity) “resulting from an unknown SCM $\mathcal{M}$, but whose graph $\mathcal{G}$ is known, a DAG, and s.t. its do-SVs are *identifiable* in $\mathcal{G}$ (i.e. all $\nu(\textbf{S})$ terms, with $\textbf{S} \subseteq \textbf{X}$, are identifiable). Note that $\mathcal{G}$ may include latent confounders as long as its do-SVs are identifiable.” This means that we do not need to assume, for example, ignorability (there may be latent confounders). As long as we have a DAG G (assumed), and this G results in $\nu(S)$ queries that are identifiable (assumed) we do not need to add any further assumptions. Depending on the shape of the graph, this do-SV identifiability assumption entails, for example, strong ignorability for some of its queries $\nu(S)$, but it is not required to assume it explicitly, as the graph itself can entail these assumptions through its structure. See also the semi-Markovian graph in Figure 3, for the experiment in section 5.1 and Appendix E.1.2, where a latent confounder appears between X and B and still results in identifiable do-SVs. SCMs are still able to compute do-SVs on this graph despite the confounder, thanks to the fact that the queries are non-parametrically identifiable (proved through do-calculus), as explained in section 3.2 and (Parafita & Vitria, 2022).
> Regarding SUTVA, as far as we know, it is not a standard assumption explicitly stated in the Causal Graphs literature (Pearl’s perspective, not Rubin’s); consistency is entailed from the SCM framework directly, while no-interference may require further assumptions. See [1] below for an in-depth discussion of the relationship between both frameworks.
>
> (Rebuttal continues in the following message)

---

> > ### Author Response · Authors · 2024-11-14
> > **Rebuttal (part 2)**
> >
> > W3: the power of do-SHAP over SHAP is not an empirical discussion, given that explanations do not (generally) have ground truth to compare against, since that ground truth is the self-same explanation we are trying to find and therefore not accesible, even on synthetic datasets. Instead, we argue this point in the paragraph mentioned in W1: provided we can estimate do-SHAP (which our experiments do test), it can be argued that non-causal SHAP cannot accurately reflect the real world mechanics due to the ignored underlying causal structure, while do-SHAP can. Nonetheless, as an illustrative example, we do include an example of marginal-SHAP estimations on the experiment in Figure 2, showing how its estimations differ significantly from the ground truth do-SHAP values that, as theoretically argued, better reflect real world mechanics.
> >
> > W4: First of all, it may be important to clarify that this paper does not propose a new explainability technique, but enhance an existing one, do-SHAP (Jung et al., 2022), which was impractical due to its dependence on estimands. Further, we make it more efficient thanks to the FRA algorithm. As such, it is not the goal of this paper to compare against other explainers. Regardless, while there is value in such a comparison, this cannot be made on equal terms (causal vs. non-causal explanations) and cannot be compared on a quantitative way (supported by ground-truth, since it is not accessible), as argued above. An empirical evaluation would only be possible by qualitatively comparing explanations, highlighting differences between explainers in particular examples with well-known causal mechanisms, which would be a subjective evaluation and prone to prioritizing plausible explanations over faithful explanations (see [2], where both qualities are discussed and it is further argued that “Faithfulness evaluation should not involve human-judgement on the quality of interpretation”). We find our experiments are better suited to our actual contributions: 1) does the estimand-agnostic approach accurately estimate do-SVs? 2) Does FRA accelerate do-SV estimation without a computational overhead? And 3) by exemplifying its applicability on real-world use cases (appendix F). In summary, do-SHAP is a pre-existing technique and as such it is not our place to continue discussing its merits nor it is feasible to do so in a quantitative manner. Instead, we focus on demonstrating our actual contributions, in making do-SHAP viable in real-world scenarios and more efficient with FRA.
> >
> > W5: we employ bold text to clarify important ideas and to clarify our contributions. However, we now see that it may hinder the readability of our work; we will rectify this stylistic choice for the camera-ready version. Thank you for pointing this out.
> >
> > [1] Ibeling, D., & Icard, T. (2024). Comparing causal frameworks: Potential outcomes, structural models, graphs, and abstractions. *Advances in Neural Information Processing Systems*, *36*.
> >
> > [2] Jacovi, A., & Goldberg, Y. (2020). Towards Faithfully Interpretable NLP Systems: How Should We Define and Evaluate Faithfulness?. In *Proceedings of the 58th Annual Meeting of the Association for Computational Linguistics*. Association for Computational Linguistics.

---

> > > ### Comment · Reviewer_98yc · 2024-11-25
> > >
> > > Thank you for the response; it cleared up some concerns. I have some follow-up questions as follows:
> > >
> > > 1/ It is mentioned that: "Note that may include latent confounders as long as its do-SVs are identifiable"
> > > =>  If latent confounders exist, why are the do-SVs identifiable? You are calculating Eq. (3), but if there is a latent confounder that is a common cause of both $S$ and $Y$, how can you calculate it?
> > >
> > > 2/ Without SUTVA: if the treatment (your $S$) of one unit affects the outcome ($Y$) of other units, how can you ensure that the estimation of $\nu_x(S)$ is unbiased?
> > >
> > > You mentioned: "No-interference may require further assumptions." -> No-interference is part of SUTVA.
> > >
> > > 3/ "provided we can estimate do-SHAP (which our experiments do test), it can be argued that non-causal SHAP cannot accurately reflect the real world mechanics due to the ignored underlying causal structure, while do-SHAP can."
> > >
> > > => Since the causal structure is known for synthetic data, why can't you illustrate the weaknesses of the other methods in your experiments?

---

> > > > ### Author Response · Authors · 2024-11-27
> > > >
> > > > Thank you for your response. We will attempt to further clarify these new questions.
> > > >
> > > > 1/ This is covered in section 3.2, where identifiability is discussed. A certain causal query (for instance, a coalition-value $\nu_{\textbf{x}}(\textbf{S}) := \mathbb{E}[Y \mid do(\textbf{S}=\textbf{s})]$) can be deemed non-parametrically identifiable or not (without explicit assumptions about the functional mechanisms $f_X$ or the prior distributions $P(\mathcal{E})$ and $P(\mathcal{U})$) by the application of the ID algorithm (Shpitser & Pearl, 2006a). This algorithm determines that the query is or is not identifiable, and if so, returns an estimand formula to estimate the value for that query using only the observational distribution $P(\mathbf{X})$. If this is the case, an SCM following the same causal structure as the underlying mechanism and trained to model the self-same $P(\mathbf{X})$ will necessarily return the correct estimation for these identifiable queries, even if the prior distributions or the functional forms are different from the underlying Data Generative Process (DGP), since these estimations depend only on the observational data distribution formed by the measured variables $\mathcal{V}$, not the confounders $\mathcal{U}$ or the exogenous noise signals $\mathcal{E}$. Given that both the underlying DGP and our trained SCM have the same causal structure and observational distribution, it can only be that the estimations for non-parametrically identifiable causal queries output the same result; otherwise, these queries would not be identifiable. Please see section 3.2 or (Parafita & Vitrià, 2022) for further elaboration on this topic.
> > > >
> > > > In summary, even with latent confounders with unknown priors or without knowledge about the functional form of the SCM, a causal query can be estimated with SCMs provided the query is non-parametrically identifiable (which can be resolved by the application of the ID algorithm on the causal structure alone) and as long as the SCM correctly models the data distribution $P(\mathbf{X})$ (which is why modeling capacity is essential to guarantee correct estimations of the query).
> > > >
> > > > 2/ See Brady Neal's "Introduction to Causal Inference" textbook (https://www.bradyneal.com/causal-inference-course#course-textbook), page 14, where SUTVA is stated as the combination of the consistency and no-interference assumptions. This is the reason why both were mentioned in our rebuttal as part of SUTVA.
> > > >
> > > > As far as we know, the Structural Causal Model framework entails both of these assumptions by construction, which is why they were not added as assumptions of our work, as they are considered implicitly. We further pointed to Ibeling's work ([1] in our rebuttal) for a discussion on the relationship between the Potential Outcomes framework and the SCM framework.
> > > >
> > > > 3/ We have included one such experiment in the new version of the paper; see the new Appendix G.
> > > >
> > > > We kindly ask you to reconsider your scores in light of our revisions and detailed rebuttals to support the publication of our work. Thank you for your time and consideration.

---

> > > > > ### Comment · Reviewer_98yc · 2024-12-02
> > > > >
> > > > > Thank you for your further clarification. Based on your responses and the new version, a critical requirement is that the method would need a true causal structure as input. In real life, identifying the causal structure is a critical problem, and in many cases, it is unidentifiable--for example, when the data consists of Gaussian noise. Hence, we may only be able to know part of the 'true' causal structure. Would this affect the explainability of the proposed method?

---

> > > > > > ### Author Response · Authors · 2024-12-02
> > > > > >
> > > > > > Thank you for your response.
> > > > > >
> > > > > > Knowing the causal graph is both an assumption of this work and a recognized limitation, as outlined in the paper (Assumption 4.1 and Section 4.4). However, it is important to note that this is a standard assumption in the SCM literature, not unique to our work.
> > > > > >
> > > > > > The paper also discusses ways to address this limitation. Domain experts can provide best-guess graphs informed by their experience and experimental knowledge of the domain. Additionally, Causal Discovery techniques, an emerging field, aim to infer domain structure from observational and/or interventional data. While these alternatives have limitations and leave room for improvement, exploring these directions is beyond the scope of this work.
> > > > > >
> > > > > > Another critical aspect is the sensitivity of estimation to graph misspecification. Further research on this topic, particularly efforts toward achieving doubly-robust guarantees for SCMs, could significantly enhance our approach. This has been identified as a valuable direction for future work.
> > > > > >
> > > > > > Finally, despite these challenges, the core argument remains: without an estimand-agnostic approach, do-Shapley explanations are generally impractical. Our work is instrumental in addressing this by making such explanations feasible and more efficient through FRA enhancements.
> > > > > >
> > > > > > We trust this addresses your concerns and thank you for engaging in this discussion.

---

### Author Response · Authors · 2024-11-27
**Revision summary**

To all reviewers,

As requested, we have included all promised changes to a new version of the paper, now uploaded. In particular:

- We have included a new section, on Appendix G, comparing do-SHAP against non-causal alternatives (marginal-SHAP and conditional-SHAP). This new section extends the argument presented in our rebuttals with a new synthetic experiment illustrating the differences in estimation when employing each of the methods, and how non-causal approaches misrepresent the underlying data generating process, hence resulting in unreliable explanations.
- We have included a new section, on Appendix H, illustrating the proposed method (estimand-agnostic do-SHAP estimation with FRA) step by step. The focus on this section is on giving a complete overview of the method from start to finish, as a way to facilitate understanding for new readers.
- We have rectified on our overuse of bold-face text. We hope this helps the readability of our work.
- We have rectified the abstract to avoid misrepresenting our contribution, as pointed out by reviewer DDJF.
- We have re-introduced Deep Causal Graphs (abbreviated as DCG) on line 442, as it was previously unknowingly omitted.
- We have corrected a case of double negative on footnote 6.

We kindly request reviewers to reconsider their scores in light of our comprehensive rebuttals and the substantial improvements made. Your support would help bring this work to the attention of the explainability community. Thank you for your time and consideration.

---

### Meta-Review · Area_Chair_wZDJ · 2024-12-23

**Metareview:**

The authors propose the Frontier Reducibility Algorithm (FRA) to make the computation of Do-Shapley values more efficient in the presence of causal structure. The final reviews are borderline, with the most positive reviewer acknowledging during discussion that the paper could stand to receive another round of review. Concerns were also raised about the lack of a timely revision.

**Additional Comments On Reviewer Discussion:**

All three reviewers acknowledged the response, and noted that the authors did not upload a revision. It appears as though the authors did finally upload a revision after reviewers commented on this, however, in the future it is recommended that this is uploaded with the response. The revision also does not make it easy to identify changes between versions.

---

### Decision · Program_Chairs · 2025-01-22

Reject